# ROTATED RUNTIME SMOOTH: TRAINING-FREE ACTIVATION SMOOTHER FOR ACCURATE INT4 INFERENCE

**Ke Yi**[1,2*], **Zengke Liu**[3,4*], **Jianwei Zhang** [2], **Chengyuan Li**[2],
**Tong Zhang**[1†], **Junyang Lin**[2],**Jingren Zhou**[2†]

## ABSTRACT

Large language models have demonstrated promising capabilities upon scaling up parameters. However, serving large language models incurs substantial computation and memory movement costs due to their large scale. Quantization methods have been employed to reduce service costs and latency. Nevertheless, outliers in activations hinder the development of INT4 weight-activation quantization. Existing approaches separate outliers and normal values into two matrices or migrate outliers from activations to weights, suffering from high latency or accuracy degradation. Based on observing activations from large language models, outliers can be classified into channel-wise and spike outliers. In this work, we propose Rotated Runtime Smooth (**RRS**), a plug-and-play activation smoother for quantization, consisting of Runtime Smooth and the Rotation operation. Runtime Smooth (**RS**) is introduced to eliminate **channel-wise outliers** by smoothing activations with channel-wise maximums during runtime. The Rotation operation can narrow the gap between **spike outliers** and normal values, alleviating the effect of victims caused by channel-wise smoothing. The proposed method outperforms the state-of-the-art method in the LLaMA and Qwen families and improves WikiText-2 perplexity from 57.33 to 6.66 for INT4 inference.

## 1 INTRODUCTION

Large language models have demonstrated promising capabilities as parameters are scaled up. However, serving large language models is plagued by the high cost of computation and memory movement due to their scale. Consequently, many quantization methods are applied to reduce size and gain throughput improvement. From the service perspective, quantization can be categorized as weight-only quantization and weight-activation quantization. The former can focus on compressing the model's weights and saving costs related to memory movement, which is suitable for the memory-bound decoding stage. The latter quantizes both weight and activation to low bits and utilizes low-bit matrix multiplication kernels to achieve speedup. However, the existence of outliers in activation stretches the quantization range, compressing the effective bits for normal values and thus hindering the development of low-bit weight-activation quantization.

To address outliers, previous works such as (Kim et al., 2023; Dettmers et al., 2022) separate outlier and normal values into two matrices. However, the implementation is not hardware-compatible and fails to expedite inference. To achieve acceleration and maintain accuracy under A8W8 quantization, SmoothQuant (Xiao et al., 2023) transfers appropriate outliers from activation to weight offline through channel-wise smoothing scales. Nevertheless, the offline smoothing scales would be ineffective when facing unmatched input, and the outlier sharing scheme makes weight difficult to quantify. The aforementioned reason impedes the implementation of SmoothQuant for A4W4 quantization. QuaRot (Ashkboos et al., 2024) utilizes the property that rotation can suppress outliers;

---

Work done during the internship in Qwen Group. * denotes equal contribution. † denotes equal advising.
[1] South China Univesity of Technology  [2] Alibaba group  [3] Key Laboratory of System Software (CAS) and State Key Laboratory of Computer Science, Institute of Software, Chinese Academy of Sciences  [4] University of Chinese Academy of Sciences. Project page: `https://coco58323.github.io/rrs2024.github.io/`.

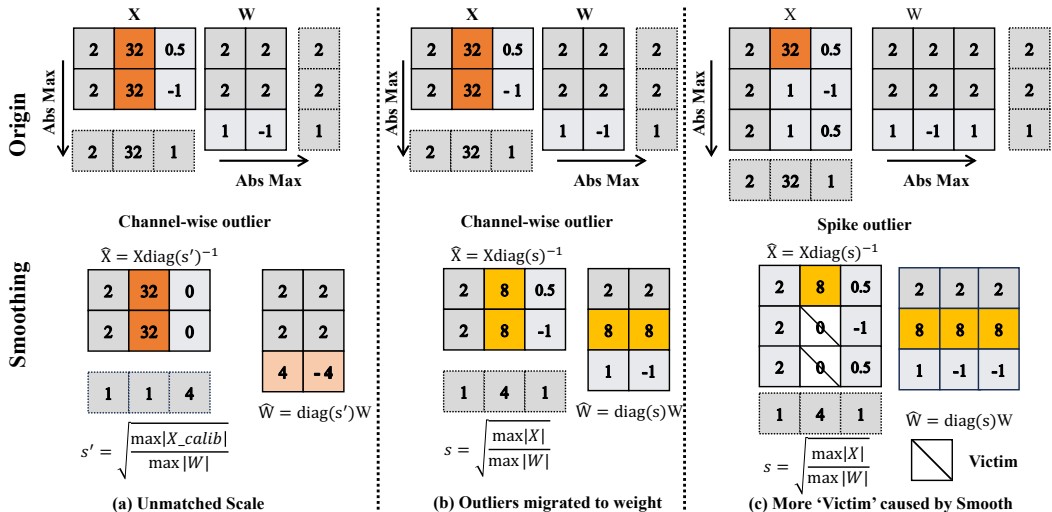

Figure 1: Challenges of SmoothQuant faced with outliers. (a) The scale $s'$, which does not match the channel-wise maximums of activations, is ineffective for smoothing purposes. (b) The migration scheme makes it difficult to quantize the smoothed activation/weight down to 4 bits. (c) Normal values are pruned as **victims** after smoothing due to the spike outlier. Note that only calibration but no quantization is involved in the above process.

hence, pairwise rotate the activation and weight with equivalent output. The rotated activation and weight tend to spread outliers internally, leading to effectiveness smoothness for A4W4 quantization. However, rotation cannot guarantee a smoother matrix, and a rotated matrix may still exhibit the shape of channel-wise outliers, as depicted in Figure 2. Therefore, deriving a more robust, accurate, and training-free scheme for INT4 inference remains an open challenge.

Based on the observation, outliers could be categorized into channel-wise outliers and spike outliers, as shown in Figure 1. To handle **channel-wise outliers**, we propose **Runtime Smooth**. Firstly, weights and activations are reordered to gather outliers and normal values. Subsequently, we group up activations and smooth activations by dividing group-wise maximums. The later quantized smoothed activation, weight, and group-wise maximums are inputs for the fused GEMM kernel. The entire process incurs minimal overhead compared to the original A4W4 pipeline. However, the existence of spike outliers causes the effect of victims after channel-wise smoothing, as shown in Figure 1. To address both **channel-wise outliers** and **spike outliers**, we propose **Rotated Runtime Smooth**, where we rotate weights and activations following (Ashkboos et al., 2024) and apply Runtime Smooth on rotated activations. The spike outlier is spread along with its token, leading to a smoother token with consistent values. The consistent values are comparatively larger than the normal values, thereby serving as smoothing scales for Runtime Smooth. The genesis of victims is abnormal smoothing scales, and the consistent smoothing scales across channels prevent the existence of victims.

To evaluate the proposed method, we conducted experiments on LLaMA families, Qwen families, Mistral, etc. On LLaMA3-70B, Rotated Runtime Smooth can gain perplexity improvement from 57.33 to 6.66 under A4W4 quantization compared with the state-of-the-art. We summarize our contributions as follows:

- We comprehensively revisited the activation smoothing method for LLM quantization, concluding the reasons for success or failure under A4W4 quantization.

- We propose Runtime Smooth, a plug-and-play component that eliminates channel-wise outliers of activation in runtime without migrating outliers to weights, bringing negligible overhead for INT4 matrix multiplication.

- We propose Rotated Runtime Smooth to overcome the spike outliers and enhance the robustness for channel-wise outliers. A comprehensive evaluation validates the effectiveness of the proposed methods, which gain thorough improvement on various models for INT4 inference.

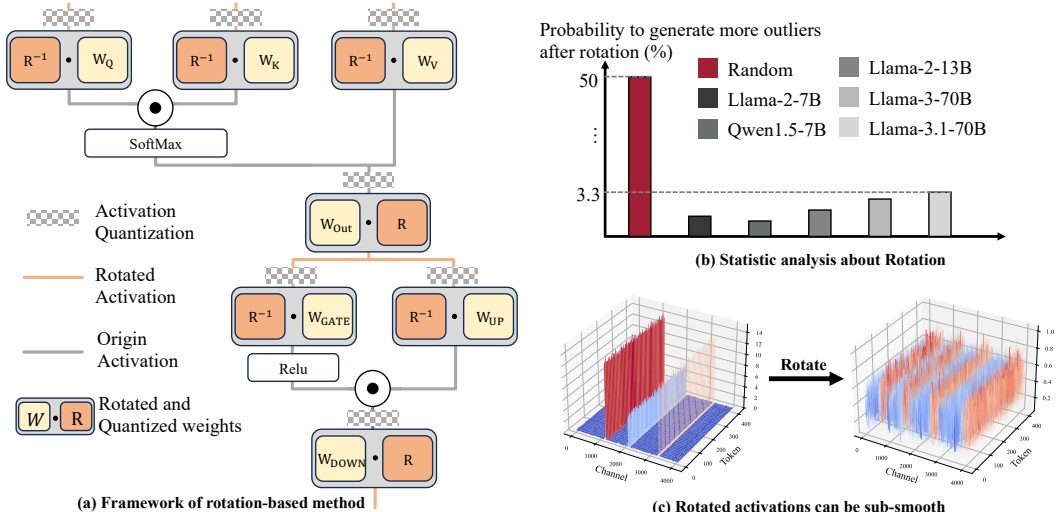

Figure 2: Review of the rotation-based method. (a) illustrates a simple implementation of the rotation-based method. The output from projector is not changed since $\mathbf{Y} = (\mathbf{XR})(\mathbf{R}^{-1}\mathbf{W}^T) = \mathbf{XW}^T$. (b) explains the success of the rotation-based method on LLMs, where activations have high confidentiality to be smoothed after rotation compared with a random matrix. (c) illustrates that activation with channel-wise outliers still maintains sub-smooth after rotation, leaving space for further smoothing

## 2 PRELIMINARIES

### 2.1 QUANTIZATION

Quantization converts high-precision matrices into discrete elements with scaling factors, achieving a lower bit per element. The process of quantization can be expressed as $\mathbf{X_{INT}} = \lfloor \frac{\mathbf{X}}{\alpha} \rceil, \alpha = \frac{\max(|\mathbf{X}|)}{2^{N-1}-1}$, where $\mathbf{X}$ represents the floating-point tensor and $\alpha$ is the scaling factor. The existence of outliers stretches the scaling factor and leaves few effective bits for normal values. Dividing the matrix into groups for quantization can mitigate the effect of outliers. In previous literature, the **per-tensor** quantization considers the entire matrix as a group; the **per-channel** quantization assigns different scaling factors to each row, and the **sub-channel** quantization divides rows into fine-grained groups. Although fine-grained grouping can alleviate accuracy degradation, more scaling factors entail additional computation and storage costs. In this work, we adopt the **per-channel** scheme following (Xiao et al., 2023; Ashkboos et al., 2024; Liu et al., 2024) for INT4 quantization.

### 2.2 CHANNEL-WISE SMOOTHING METHOD

Under the assumption that outliers persist in fixed channels of activations, SmoothQuant migrates the outliers by dividing the smoothing scale $\mathbf{s} \in \mathbb{R}^K$, where k denotes channel dimension. For ensuring equivalence of output, $\mathbf{s}$ would be multiplied to weight; the process can be described as $\mathbf{Y} = (\mathbf{X}\mathrm{diag}(\mathbf{s})^{-1})(\mathrm{diag}(\mathbf{s})\mathbf{W}^T) = \hat{\mathbf{X}}\hat{\mathbf{W}}^T$. The smoothing scales are computed as $\mathbf{s}_j = \max(|\mathbf{X}_j|)^\alpha / \max(|\mathbf{W}_j|)^{1-\alpha}, j = 1, 2, \ldots, K$, to fairly share the outliers between weights and activations. Since weights are mostly quantized offline (Frantar et al., 2022), directly multiplying $s$ during runtime would undermine the quantization property. Therefore, $s$ is pre-computed using a calibration set and merged into weights before quantization.

Although SmoothQuant is effective under the A8W8 scheme, it fails for INT4 inference in three respects, as shown in Figure 1. Firstly, the smoothing scales depending on the calibration set are prone to being unmatched with online activations; hence, they cannot smooth outliers. Secondly, the outlier channels of activation are not eliminated but partially migrated to weights, thus leading to failure in low-bit quantization. Thirdly, outliers are not always channel-wise consistent, where spike outliers exist, and normal values are pruned as 'victim' after smoothing.

**Spike Outliers and Effect of Victim**: Spike outliers stretch the smoothing scale. The normal values divided by abnormal smoothing scales are minimal compared to other elements in the quantization group; hence, they become 'victims'. The existence of the victims leads to quantization error.

## 2.3 ROTATION-BASED METHOD

A rotation matrix is an orthogonal matrix R satisfied $\mathbf{R}\mathbf{R}^T = \mathbf{I}$ and $det(\mathbf{R}) = 1$. Quip (Tseng et al., 2024) showed that multiplying a weight matrix on the left and right by an orthogonal matrix can theoretically alleviate outliers, making matrices easier to quantize. QuaRot (Ashkboos et al., 2024), employs a similar technique by multiplying weight or activations by only one rotation matrix, maintaining an equivalent output as depicted in Figure 2 (a). However, multiplying one rotation matrix could not theoretically guarantee a smoother weight or activation. To explain the success of the rotation-based method, we computed the probability that the activation of different models becomes less smooth after rotation, as illustrated in Figure 2 (b). Following previous works, we measure the smoothness as $\mu = \text{abs}(\max(\mathbf{t}))/\text{RMS}(\mathbf{t})$, where $\mathbf{t}$ denotes one token in activation, and RMS denotes root mean square. Rotating activation from LLMs consistently exhibits a low probability of being less smooth compared with rotating a random matrix. However, having a chance to be less smooth is a potential trouble. The appendix provides a detailed explanation of probability calculation. On the other hand, activations with channel-wise outliers can be viewed as a collection of vectors with the same direction. From the rotation property, the rotated activation might be sub-smooth, as shown in Figure 2 (c), leaving space for better smoothing.

## 3 METHODOLOGY

In this section, we introduce Runtime Smooth to eliminate channel-wise outliers (3.1) and how to implement it efficiently with kernel fusion (3.2). To comprehensively eliminate outliers, we propose Rotated Runtime Smooth, which addresses the effect of the victim and sub-smoothness of rotated activation 3.3.

## 3.1 RUNTIME SMOOTH

Challenges for the smoothing-based method under INT4 inference are discussed in Section 2.2. One intuitive way to mitigate this challenge is to obtain smoothing scale $s$ in runtime and not merge $s$ into weights. The process can be formulated as:

$$\mathbf{s}_j = \max(|\mathbf{X}_j|), j = 1, 2, ..., K$$

$$\overline{\mathbf{X}} = \text{Quantize}(\mathbf{X}\text{diag}(\mathbf{s})^{-1}), \overline{\mathbf{W}} = \text{Quantize}(\mathbf{W}),$$

$$\mathbf{Y} = \sum_{j=1}^{K} \overline{\mathbf{X_j}} \cdot \overline{\mathbf{W_j}}^T \cdot \mathbf{s}_j,$$

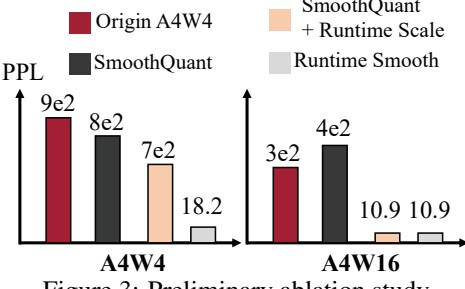

Figure 3: Preliminary ablation study

where $\mathbf{s} \in \mathbf{R}^{1 \times K}$ denotes the runtime smoothing scale, $\mathbf{X} \in \mathbf{R}^{N \times K}$ denotes activations, $\mathbf{W} \in \mathbf{R}^{M \times K}$ denotes weights, $\mathbf{X}_j \in \mathbf{R}^{N \times 1}$ denotes one column of activation. We conducted an ablation study with LLaMA3-8B and the WikiText-2 dataset to understand better the effect of unmatched scale and outlier shared scheme. As shown in Figure 3, merely applying the runtime smoothing scale could not make A4W4 feasible, whereas Runtime Smooth does, echoing the importance of not migrating outliers to weights. To avoid the effect of quantization error from weight, we further apply the A4W16 setting. The perplexity improvement, from 4e2 to 10.9, validates the effectiveness of adopting the runtime scale.

## 3.2 RUNTIME SMOOTH WITH KERNEL FUSION

However, the naive implementation cannot integrated into the GEMM pipeline. A GEMM kernel splits the input matrix into blocks by columns and conducts parallel block computation, where the inconsistent smoothing scale would make the block-wise computation complex. Intuitively, if the

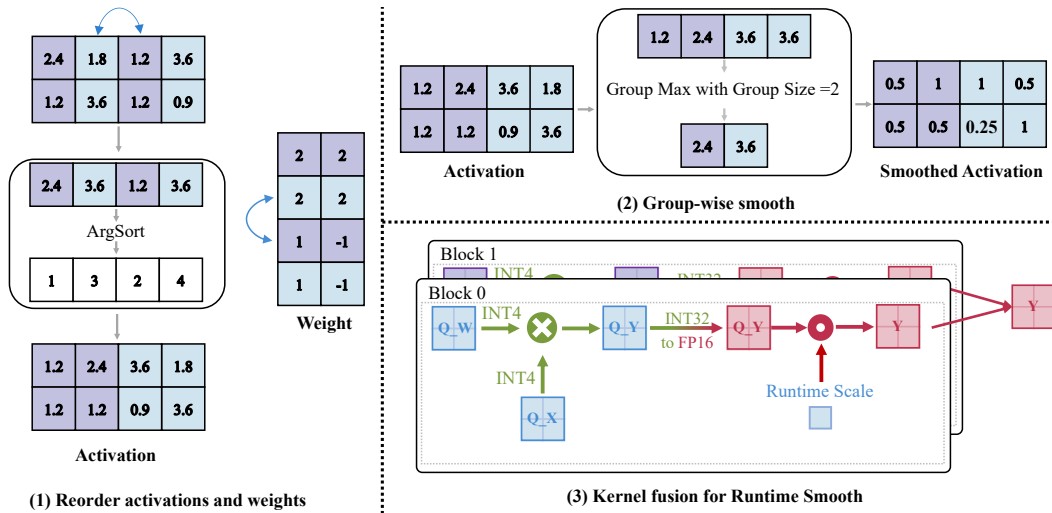

Figure 4: Pipeline of Runtime Smooth. (1) Reorder activations and weight according to channel-wise maximums of activation. Note that the reordering process would not change the final result since $\mathbf{Y} = \sum_{i=1}^{\mathbf{K}} \mathbf{X}_i @ \mathbf{W}_i^T$, and $\mathbf{Y}$ is irrelevant with the order of i. (2) Group up activations according to block size of matrix multiplication computation. The maximums of the group are set to the runtime smoothing scale of the group. (3) In the matrix multiplication pipeline, quantized smoothed activations and weights are segmented into blocks. The block size is equivalent to the previous group size. Within a block, tiled smoothed activations are multiplied by tiled quantized weights. The runtime smoothing scales are applied to the interim result.

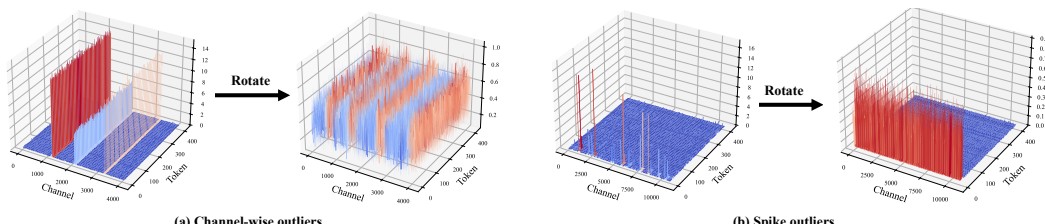

Figure 5: Analysis of rotated activations with different outliers. (a) Activation with channel-wise outliers maintains channel-wise consistency after rotation, hence being sub-smooth for quantization. (b) One spike outlier is spread on its token internal, where the smoothing scale is consistent without abnormal value, further preventing 'victim'.

smoothing scale is the same within a block, equation 3.1 can be deduced to $\mathbf{Y} = \mathbf{s} \cdot \sum_{j=1}^{K} \hat{\mathbf{X}}_\mathbf{j} \cdot \hat{\mathbf{W}}_\mathbf{j}^T$. The overhead would be minimized due to fewer multiplication operations. Roughly altering the smoothing scales to be consistent would degrade the level of smoothness. Hence, we reorder the activations and weights according to the magnitude of smoothing scales, validated as hardware-friendly with negligible cost (Lin et al., 2024b). The reordered activations group up the outliers and normal values. The group size is configured to be identical to the block size. The maximum value of an activation group is set as the smoothing scale for the corresponding computation block.

As shown in Figure 4, the pipeline of GEMM fused with Runtime Smooth can be described as 1. Reorder the activation and weight according to channel-wise maximums of activation; 2. Group up the activation and set the group-wise maximum as smoothing scales; 3. Calculate the matrix multiplication of the tiled block and multiply the runtime scale on the dequantized result, followed by a reduction operation. It is worth noting that, compared with the A4W4 baseline, the extra runtime scale multiplication only brings negligible overhead, which will be discussed in Section 4.5.

## 3.3 ROTATED RUNTIME SMOOTH

In this section, we propose Rotated Runtime Smooth (RRS) to comprehensively eliminate outliers, including both channel-wise and spike outliers. Here, the activations are rotated and subsequently applied Runtime Smooth.

For activations with channel-wise outliers, it maintains the channel-wise consistency after rotation due to the rotation property, as shown in Figure 5. Moreover, the rotated activations have a chance to raise the level of outliers with a small probability. Compared with the pure rotation method, RRS comprehensively smooths channel-wise, leading to low quantization error. Compared with Runtime Smooth, RRS is more robust to activations with tiny spikes in the practical scenario.

As discussed in Section 2.2, the existence of spike outlier is the bottleneck of Runtime Smooth. Figure 5 shows that the spike outliers are spread within tokens after rotation. Hence, the smoothing scales are more consistent across channels, leaving fewer victims since all elements are divided by such a consistent scale. The process can be described as:

$$\mathbf{R} = \frac{1}{\sqrt{K}}[c_{i,j}]_{K \times K}, \mathbf{t} = [\varepsilon, \cdots, \varepsilon, O_i, \varepsilon, \cdots, \varepsilon],$$

$$\mathbf{t}_{rotation} = \mathbf{t} \cdot \mathbf{R} = \frac{1}{\sqrt{K}}[c_{i,1}O_i + \varepsilon, c_{i,2}O_i + \varepsilon, \cdots, c_{i,K}O_i + \varepsilon], \qquad (1)$$

$$\text{smooth\_scale} = \max(|\mathbf{t}_{rotation}|) \approx \frac{1}{\sqrt{K}}[|O_i|, |O_i|, \cdots, |O_i|],$$

where the $O$ denotes spike outliers and $\varepsilon$ denotes normal values. $c_{i,j} \in \{-1, +1\}$ denotes the elements in Hadamard rotation matrix. In a practical scenario, the normal values $\varepsilon$ are relatively small compared with spike outliers $O$; hence, they are omitted in the smoothing scale as shown in Equation 1. The genesis of victims is the abnormal smoothing scale; hence a more consistent smoothing scale brought by RRS frees 'victims'. Here, we only discuss the scenario where there is only one spike in the token, and we conduct a comprehensive analysis of multiple spikes for real scenarios in Section C.1.

We apply RRS to Linear Layers in Transformer blocks. We first offline rotate the weight matrix and insert online rotation operation before output and down projectors following previous works (Ashkboos et al., 2024). The rotated weights are quantized offline with GPTQ (Frantar et al., 2022). During inference, we perform Runtime Smooth on the rotated activations of linear layers and apply activation quantization subsequently.

# 4 EXPERIMENTS

## 4.1 SETTINGS

We conduct experiments on mainstream LLMs, including LLaMA families (Touvron et al., 2023b) (LLaMA2-13B, LLaMA2-70B, LLaMA3-8B, LLaMA3-70B, LLaMA3.1-8B, LLaMA3.1-70B), Qwen families (Yang et al., 2024) (Qwen1.5-7B, Qwen1.5-14B), Mistral (Jiang et al., 2023) and Mixtral (Jiang et al., 2024). Activation quantization employs per-channel symmetric scheme with round-to-nearest (RTN) strategy. KV cache quantization employs sub-channel symmetric scheme with groupsize 128 and round-to-nearest (RTN) strategy. In most cases, weight quantization employs per-channel symmetric scheme with GPTQ (Frantar et al., 2022) strategy, except for baseline 'RTN'. We apply standard GPTQ settings by using 128 samples from WikiText-2 with a sequence length of 2048 as the calibration set. We evaluate the performance of the models on WikiText-2 perplexity and zero-shot Common Sense QA benchmarks. The Common Sense QA benchmarks include ARC-e, ARC-c (Clark et al., 2018), BoolQ (Clark et al., 2019), and OBQA (Mihaylov et al., 2018).

## 4.2 MAIN RESULT

Runtime Smooth emphasizes activation smoothing for INT4 inference. The plug-and-play Runtime Smooth operators are employed before activation quantization. We conduct a comparison between Runtime Smooth and SmoothQuant (Xiao et al., 2023), demonstrating promising improvement. For instance LLaMA2-70B: 1e2 -> 6.95, LLaMA3-8B: 8e2 -> 10.47, Qwen1.5-7B: 3e2 ->13.32 under the A4W4KV16 scheme, as shown in Table 1. For certain models like LLaMA3-70B and LLaMA3.1-70B, both Runtime Smooth and SmoothQuant fail for the difficulty of weight quantization. To eliminate the influence of quantization error from weights, we carry out experiments under A4W16KV16 settings. Under the activation-only quantization setting, Runtime Smooth consistently outperforms SmoothQuant and achieves 40x improvement on LLaMA3-8B, validating the effectiveness of Runtime Smooth. Here the group size of the smoothing scale is 1 to observe the upper bound performance.

Table 1: Comparison on WikiText-2 perplexity. We evaluate models and methods on three quantization schemes: A4W4KV4, A4W4KV16, and A4W16KV16. Results for SmoothQuant, GPTQ, and QuaRot were obtained by re-implementation based on their publicly released codebase.

| #Bits W-A-KV | Method | LLaMA 2-13B | LLaMA 2-70B | LLaMA 3-8B | LLaMA 3.1-8B | LLaMA 3-70B | LLaMA 3.1-70B | Mixtral | Mistral | Qwen 1.5-7B | Qwen 1.5-14B |
|---|---|---|---|---|---|---|---|---|---|---|---|
| 16-16-16 | FP16 | 5.00 | 3.30 | 6.13 | 6.24 | 2.80 | 2.81 | 3.84 | 5.94 | 7.95 | 7.44 |
| 16-4-16 | RTN | 5e3 | Nan | 4e2 | 2e2 | 3e4 | 1e4 | 8e2 | 7e2 | 6e3 | 2e4 |
| | SmoothQuant | 97.04 | Nan | 5e2 | 2e2 | 74.23 | 48.59 | 88.68 | 47.92 | 2e2 | 1e2 |
| | **RS** | 6.40 | 4.52 | 11.44 | 9.71 | 11.31 | 7.27 | 5.72 | 7.39 | 10.95 | 9.74 |
| | QuaRot | 5.24 | **3.72** | 7.77 | 7.81 | 1e2 | 5.67 | 4.68 | 6.29 | 9.07 | 8.18 |
| | **RRS** | **5.22** | 3.74 | **7.55** | **7.60** | **6.32** | **5.03** | **4.50** | **6.21** | **8.91** | **8.16** |
| 4-4-16 | RTN | 7e3 | 2e5 | 9e2 | 5e2 | 1e5 | 2e4 | 8e2 | 7e2 | 9e3 | 3e4 |
| | SmoothQuant | 34.50 | 1e2 | 8e2 | 4e2 | 5e2 | 2e2 | 3e2 | 76.25 | 3e2 | 1e2 |
| | GPTQ | 5e3 | 2e6 | 9e2 | 4e2 | 1e5 | 2e4 | 1e3 | 6e2 | 1e4 | 3e4 |
| | **RS** | 8.79 | 6.95 | 10.47 | 10.39 | 7e3 | 1e4 | 7.37 | 8.16 | 13.32 | 10.38 |
| | QuaRot | 5.39 | **3.85** | 8.38 | 8.38 | 57.33 | 6.26 | 4.80 | 6.38 | 9.34 | 8.32 |
| | **RRS** | **5.36** | 3.86 | **8.11** | **8.12** | **6.66** | **5.56** | **4.63** | **6.31** | **9.17** | **8.29** |
| 4-4-4 | RTN | 7e3 | 2e5 | 1e3 | 6e2 | 1e5 | 2e4 | 9e2 | 7e2 | 1e4 | 2e4 |
| | SmoothQuant | 56.60 | 1e2 | 1e3 | 6e2 | 5e2 | 3e2 | 2e2 | 78.39 | 3e2 | 2e2 |
| | GPTQ | 5e3 | 1e6 | 1e3 | 5e2 | 1e5 | 2e4 | 1e3 | 6e2 | 1e4 | 3e4 |
| | **RS** | 9.15 | 7.08 | 11.52 | 11.19 | 8e3 | 1e4 | 7.98 | 8.64 | 13.97 | 10.62 |
| | QuaRot | 5.51 | **3.89** | 8.76 | 8.80 | 49.73 | 6.46 | 4.93 | 6.45 | 9.55 | 8.43 |
| | **RRS** | **5.45** | **3.89** | **8.42** | **8.49** | **6.87** | **5.69** | **4.74** | **6.35** | **9.37** | **8.35** |

To further narrow the accuracy gap between INT4 inference and full precision inference, we propose Rotated Runtime Smooth. Here, the group size of the smoothing scale is set to 128, which is identical to the GEMM block size, enabling efficient implementation. Compared with the state-of-the-art, QuaRot, Rotated Runtime Smooth consistently outperforms across different model sizes and model families. In most instances, Rotated Runtime Smooth achieves a 0.1 - 0.3 improvement in WikiText-2 perplexity. It is noteworthy that on LLaMA3-70B, Rotated Runtime Smooth reduces perplexity from 57.33 to 6.66 under the A4W4KV16 setting and from 49.76 to 6.87 under the A4W4KV4 setting. Notably, 57.33 and 49.76 are abnormal results. Hence, we analyze outliers from rotated LLaMA3-70B and LLaMA3.1-70B. The distribution of outliers is similar for the two models. More details are provided in Section C.2. This indicates that models have different sensitivities to outliers. Further suppressing the outliers can enhance robustness.

We also conduct a comparison between the proposed approaches and the state-of-the-art methods on the zero-shot common sense QA task, as presented in Table 2. The Rotated Runtime Smooth consistently surpasses the baseline by approximately 3% in terms of average accuracy improvement. In certain cases, LLaMA3-8B and LLaMA3.1-8B, Runtime Smooth outperforms QuaRot. The latter requires complex online Hadamard rotation as described in (Ashkboos et al., 2024). More results are listed in the Appendix A

## 4.3 COMPARISON WITH TRAINING-BASED METHOD

SpinQuant (Liu et al., 2024) suggests that diverse Rotation matrices exhibit variations in their impact of smoothing. Consequently, it substitutes the origin fix Rotation matrix with a trainable Rotation matrix and trains the rotated network. The training process is time-consuming, taking 1.5 hours for a 7B model on one A100 GPU and 12 hours for 70B models on eight A100 GPUs. We re-implement SpinQuant and compare it with our method as shown in Table 3. It is noteworthy that SpinQuant applies asymmetric quantization to activation and KV cache. We experiment with our method under the same settings. The result reveals that the training-based method degrades WikiText-2 perplexity compared to the training-free method and still has room for improvement.

Table 2: 0-shot accuracy (%) on the Common Sense QA tasks. Each block is based on the same foundation model specified in the row. We organize all results under different quantization schemes.

| #Bits | Model | Method | OBQA | BoolQ | ARC_E | ARC_C | Avg. |
|---|---|---|---|---|---|---|---|
| 4-4-16 | LLaMA-3-8B | GPTQ | 26.6 | 44.0 | 29.8 | 21.0 | 30.4 |
| | | SmoothQuant | 25.0 | 43.7 | 30.2 | 22.9 | 30.4 |
| | | **RS** | 42.4 | 65.6 | 72.2 | 46.3 | 56.6 |
| | | QuaRot | 39.2 | 70.7 | 68.6 | 41.5 | 55.0 |
| | | **RRS** | 42.4 | 73.6 | 71.1 | 44.8 | **58.0** |
| | LLaMA-3.1-8B | GPTQ | 26.2 | 44.7 | 30.3 | 23.9 | 31.3 |
| | | SmoothQuant | 26.2 | 46.42 | 30.9 | 24.5 | 32.0 |
| | | **RS** | 44.6 | 68.8 | 72.5 | 47.6 | 58.4 |
| | | QuaRot | 37.6 | 75.6 | 72.9 | 44.8 | 57.7 |
| | | **RRS** | 42.4 | 78.1 | 76.1 | 50.3 | **61.7** |
| | Mistral | GPTQ | 27.0 | 46.1 | 32.0 | 25.0 | 32.5 |
| | | SmoothQuant | 28.6 | 57.5 | 42.2 | 31.4 | 39.9 |
| | | **RS** | 35.2 | 80.2 | 72.4 | 51.1 | 59.7 |
| | | QuaRot | 44.4 | 83.1 | 71.9 | 54.4 | 63.4 |
| | | **RRS** | 43.4 | 85.1 | 74.4 | 55.4 | **64.6** |
| | Qwen1.5-7B | GPTQ | 28.2 | 42.9 | 25.9 | 26.5 | 30.9 |
| | | SmoothQuant | 28.2 | 54.0 | 32.3 | 26.7 | 35.3 |
| | | **RS** | 37.6 | 72.6 | 56.9 | 39.0 | 51.5 |
| | | QuaRot | 39.0 | 73.9 | 61.3 | 40.4 | 53.7 |
| | | **RRS** | 43.0 | 77.7 | 61.5 | 42.0 | **56.0** |
| 4-4-4 | LLaMA-3-8B | GPTQ | 28.2 | 42.0 | 28.8 | 24.1 | 30.7 |
| | | SmoothQuant | 28.8 | 52.2 | 37.6 | 28.8 | 36.9 |
| | | **RS** | 43.2 | 65.7 | 67.2 | 41.5 | 54.4 |
| | | QuaRot | 38.6 | 70.6 | 66.7 | 40.4 | 54.1 |
| | | **RRS** | 42.4 | 73.5 | 68.7 | 44.7 | **57.3** |
| | LLaMA-3.1-8B | GPTQ | 28.0 | 43.6 | 29.0 | 23.3 | 31.0 |
| | | SmoothQuant | 28.0 | 45.1 | 30.0 | 22.7 | 31.4 |
| | | **RS** | 43.4 | 65.3 | 71.1 | 46.1 | 56.5 |
| | | QuaRot | 37.2 | 73.5 | 65.7 | 41.7 | 54.5 |
| | | **RRS** | 41.8 | 77.8 | 75.3 | 48.1 | **60.8** |
| | Mistral | GPTQ | 24.8 | 46.2 | 31.3 | 25.8 | 32.0 |
| | | SmoothQuant | 27.0 | 56.6 | 40.6 | 29.5 | 38.4 |
| | | **RS** | 36.6 | 81.0 | 72.4 | 50.3 | 60.1 |
| | | QuaRot | 43.2 | 83.0 | 72.8 | 52.5 | 62.9 |
| | | **RRS** | 45.6 | 84.3 | 73.2 | 54.6 | **64.4** |
| | Qwen1.5-7B | GPTQ | 30.0 | 42.8 | 26.2 | 26.5 | 31.4 |
| | | SmoothQuant | 28.0 | 53.1 | 30.2 | 25.1 | 34.1 |
| | | **RS** | 36.8 | 70.2 | 56.2 | 40.2 | 50.9 |
| | | QuaRot | 39.6 | 74.3 | 62.1 | 42.0 | 54.5 |
| | | **RRS** | 40.4 | 76.6 | 61.8 | 42.5 | **55.3** |

Table 3: Comparison with the training-based method, SpinQuant, where the result was obtained by re-implementation based on its publicly released codebase

| Method | LLaMA-2-7B | LLaMA-2-13B | LLaMA-3-8B | LLaMA-3.1-8B |
|---|---|---|---|---|
| Spinquant | 6.37 | 5.55 | 7.99 | 7.93 |
| QuaRot | 6.03 | 5.35 | 7.91 | 7.89 |
| RRS | 5.99 | 5.29 | 7.79 | 7.76 |

Table 4: Ablation study of Group size of runtime smooth scale

| Method | Model | 1 | 32 | 64 | 128 | 256 | 512 |
|---|---|---|---|---|---|---|---|
| RRS | Mistral | 6.29 | 6.28 | 6.30 | 6.31 | 6.31 | 6.33 |
| | Qwen1.5-7B | 9.16 | 9.14 | 9.16 | 9.17 | 9.18 | - |
| | LLaMA3.1-8B | 8.09 | 8.09 | 8.10 | 8.12 | 8.18 | 8.29 |
| RS | Mistral | 8.16 | 11.17 | 11.42 | 18.48 | 36.46 | 154.19 |
| | Qwen1.5-7B | 13.32 | 15.67 | 18.08 | 22.83 | 41.55 | - |
| | LLaMA3.1-8B | 10.39 | 13.51 | 20.74 | 51.63 | 214.88 | 2706.82 |

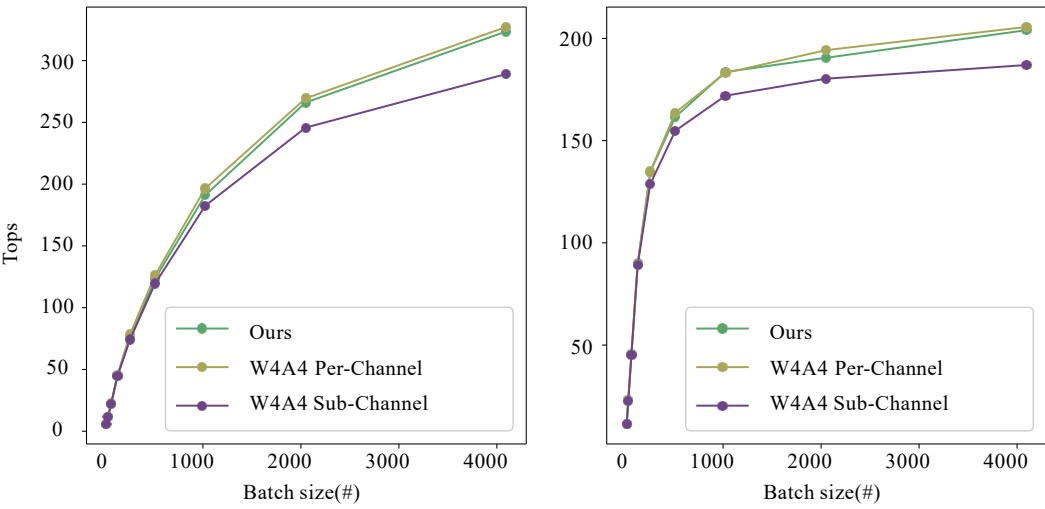

Figure 6: Performance evaluation of different quantization approaches. We set up the evaluation configuration aligned with the LLaMA-7b configuration and 1024 sequence length. Kernels are evaluated by NVBench

### 4.4 ABLATION STUDY

We conduct an ablation study on the group size of runtime smoothing scale for Mistral, Qwen1.5-7B, and LLaMA3.1-8B, as presented in 4. Runtime Smooth can effectively minimize the gap between A4W4 and full precision through a subtle group strategy. However, the accuracy deteriorates as the group size increases. Rotated Runtime Smooth employs the rotation technique to minimize the gap between outliers and normal values, thereby being robust to the coarse group scheme and enabling the implementation of the fused kernel. It should be noted that in Qwen1.5-7B, the size of the input activation for Down_projector is 11008, which does not support a group size of 512.

### 4.5 EFFICIENCY EVALUATION

We evaluate the GEMM kernel fused with Runtime Smooth on NVBench (NVIDIA, 2024) with RTX 4070 Ti, as shown in Figure 6. The group size of the smoothing scale is 128, the same as the block size of the GEMM kernel. We implement Per-Channel A4W4 and Sub-Channel A4W4 as baselines. Compared with Per-Channel A4W4, Runtime Smooth fused Kernel brings limited overhead, including the movement of the smoothing scale (shape of [1, K]) and a multiplication between matrix and scalar. Sub-Channel A4W4 brings noticeable overhead, including the movement of group-wise quantization scale (shape of [N, L] and [M, L]) and multiplication between matrices. Hence, across different batch sizes and hidden dimensions, Runtime Smooth fused Kernel brings negligible overhead compared with A4W4 Per-Channel quantization, which is also the setting of QuaRot and SpinQuant.

## 5 RELATED WORK

### 5.1 LARGE LANGUAGE MODELS

Pre-trained language models have achieved remarkable progress through scaling. Open-source large language models (LLMs) can reach up to 405 billion parameters and offer promising few-shot/zero-shot results. The mainstream LLMs (Yang et al., 2024; Touvron et al., 2023a; DeepSeek-AI et al., 2024; Jiang et al., 2023) continuously provide models with large scales and enhanced capabilities. However, serving large language models for inference becomes costly and challenging as the models expand.

### 5.2 MODEL QUANTIZATION

Quantization represents an effective approach for reducing model size and expediting inference. From a serving perspective, quantization can be classified into weight-only quantization and weight-activation quantization. Weight-only quantization compression employs low-bit representations for weight matrices, thereby saving memory movement in memory-bound scenarios, specifically the decoding stage. GPTQ (Frantar et al., 2022) used 4-bit to quantize the weight based on the approximate second-order information. AWQ (Lin et al., 2023) further advanced accuracy by preserving salient weights. QFA (Yi et al., 2024) fine-tune a supernetwork encompassing multiple mixed precision configurations and efficiently offer high-performance sub-networks for diverse scenarios. QuIP (Chee et al., 2024; Tseng et al., 2024) successfully represents weights using 2 bits via an adaptive rounding method. Weight-activation quantization can further accelerate computation by leveraging a low-bit GEMM kernel suitable for compute-bound scenarios, namely the pre-filling stage. Qserve (Lin et al., 2024b) further implement the A8W4KV4 and better accelerate. Among weight-activation quantization methods, the existence of outliers presents the most formidable problem, as it can result in substantial drops in accuracy.

### 5.3 OUTLIERS CHALLENGE

Outliers can expand the quantization range and compress the information intensity for normal values. LLM.int8() (Dettmers et al., 2022) employs mixed INT8/FP16 decomposition to handle activation outliers. Nevertheless, such an implementation results in significant latency overhead and can even be slower than FP16 inference. Subsequent work (Yuan et al., 2023) rearranges the channels to reduce the variance within one quantization group, further enhancing accuracy. Atom (Zhao et al., 2024) integrates the reorder technique and mixed INT4/INT8 precision to maintain accuracy and accelerate compared to the FP16 baseline. SmoothQuant (Xiao et al., 2023) exchanges outliers between weights and activations to find an optimal point that shares appropriate outliers in weights and activations, achieving A8W8 inference with minimal accuracy degradation. However, the outlier sharing scheme may not be suitable for the extreme A4W4 setting. To further smooth outliers, QuaRot (Ashkboos et al., 2024) pairwise rotates the activation and weight to suppress outliers and maintain output equalization, enabling INT4 inference with well-smoothed activations. SpinQuant (Liu et al., 2024) suggests substituting the origin fix Rotation matrix with a trainable Rotation matrix could improve accuracy. DuQuant (Lin et al., 2024a) involves double rotation and achieves a more smooth activation.

## 6 CONCLUSION

This work presents Rotated Runtime Smooth, a plug-and-play runtime activation smoother that facilitates INT4 inference. In Rotated Runtime Smooth, Runtime Smooth effectively eliminates channel-wise outliers. Additionally, rotation operations migrate the negative impact from spike outliers. Rotated Runtime Smooth can be easily implemented with negligible overhead and is generalized across various large language models (LLMs). Through comprehensive elimination of channel-wise and spike outliers, Rotated Runtime Smooth achieves significant enhancement compared to the prior channel-wise smoothing approach and outperforms state-of-the-art methods on diverse models for INT4 inference.

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

# A  Supplementary experiment

## A.1  More Downstream task

In this section, we incorporate additional downstream tasks such as MMLU, GSM8k (Math), C-EVAL (Chinese), and HumanEval (code), utilizing 7/8B models from Qwen2.5, Mistral, LLaMA 3, and LLaMA 3.1. The results show that our method effectively smooths the activations and works with out-of-distribution (OOD) data like math and code, making the improvements over the baselines not so small.

Table 5: 0-shot accuracy (%) on GSM8k, MMLU, C-Eval with Qwen2.5-7B, Mistral, LLaMA-3-8B, LLaMA-3.1-8B.

| Model | Method | GSM8k_flex (Math) | GSM8k_strict (Math) | MMLU | C-Eval (Chinese) |
|---|---|---|---|---|---|
| Qwen2.5-7B | RRS | 70.96 | 62.85 | 63.00 | 66.6 |
| Qwen2.5-7B | QuaRot | 62.77 | 52.99 | 62.15 | 66.1 |
| Mistral | RRS | 37.30 | 36.69 | 56.65 | 40.6 |
| Mistral | QuaRot | 31.15 | 30.62 | 55.16 | 39.3 |
| LLaMA3-8B | RRS | 31.46 | 31.38 | 53.51 | 35.5 |
| LLaMA3-8B | QuaRot | 24.71 | 24.56 | 50.85 | 31.3 |
| LLaMA3.1-8B | RRS | 36.08 | 35.93 | 54.96 | 38.8 |
| LLaMA3.1-8B | QuaRot | 28.50 | 28.27 | 51.64 | 34.1 |

Table 6: Evaluation on HumanEval with Qwen2.5-7B.

| Model | Method | HumanEval (Code) |
|---|---|---|
| Qwen2.5-7B | RRS | 60.98 |
| Qwen2.5-7B | QuaRot | 49.39 |

## A.2  More Baselines

We compare our method with two additional baselines: SpinQuant and DuQuant, as well as their variants that incorporate Runtime Smooth (RS). The results show that our runtime smooth is well generalizable and valid, demonstrating the importance of runtime smoothing for various data from different domains.

Table 7: 0-shot accuracy (%) on the Common Sense QA tasks. Each block is based on the same foundation model specified in the row. We organize all results under different quantization schemes.

| Method | GSM8k Flex | GSM8k Strict | MMLU | ARC_C | ARC_E | BoolQ | OBQA | Avg. |
|---|---|---|---|---|---|---|---|---|
| RRS | 31.4 | 31.3 | 53.5 | 44.8 | 71.1 | 73.6 | 42.4 | 57.9 |
| DuQuant | 24.4 | 24.0 | 49.4 | 45.8 | 68.7 | 71.3 | 42.2 | 57.0 |
| DuQuant+RS | 32.9 | 32.4 | 55.7 | 47.5 | 74.0 | 76.2 | 42.6 | 60.1 |
| SpinQuant | 22.9 | 22.6 | 48.4 | 43.2 | 67.3 | 72.7 | 38.8 | 55.5 |
| SpinQuant+RS | 23.1 | 23.2 | 52.2 | 45.1 | 70.6 | 73.6 | 39.4 | 57.1 |

# B  Measurement of Probability

In this section, we explain how to obtain the result in Figure 2. We collect activations token-wise for each module in the WikiText-2 test set for LLMs. In order to determine the probability of the metric, we first compute the smooth measure $\mu = \mathrm{abs}\big(\max(\mathbf{t})/\mathrm{RMS}(\mathbf{t})$, where $\mathbf{t}$ denotes one token in activation, and RMS denotes root mean square. Then we rotate the activation with the default

Hadamard matrix $R$ and recalculate $\mu' = \text{abs}\big(\max(\mathbf{t}')\big)/\text{RMS}(\mathbf{t}'), t = t@R$. Finally, we calculate the probability of $\mu < \mu'$.

## C  OUTLIER

### C.1  ACTIVATION WITH MULTIPLE SPIKE OUTLIERS

This section further analyzes how tokens with multiple outliers behave after rotation. The token with multiple outliers is defined as:

$$\mathbf{t} = [\cdots, O_{i_1}, \cdots, O_{i_2}, \cdots, O_{i_l}, \cdots], \tag{2}$$

where l denotes the number of outliers, $\{i_1, i_2, ...i_l\}$ denotes the index of outliers. In this work, we set the Hadamard matrix as the rotation matrix $\mathbf{R} = \frac{1}{\sqrt{K}}[s_{i,j}]_{K \times K}$, where $s_{i,j} \in \{-1, +1\}$. The rotated token can be described as:

$$\mathbf{t}^{rot} = \mathbf{t} \cdot \mathbf{R} \tag{3}$$

$$\approx \frac{1}{\sqrt{K}}[\sum_{d=1}^{l} s_{i_d,1} O_{i_d}, \sum_{d=1}^{l} s_{i_d,2} O_{i_d}, \cdots, \sum_{d=1}^{l} s_{i_d,K} O_{i_d}]. \tag{4}$$

The construction of $t^{rot}$ can be viewed as contributions of outliers, where outliers are canceled out by each other or stacked to enlarge. The effect of victims refers to the smoothness of normal tokens after smoothing. The process is defined as:

$$\mathbf{x} = [1, 1, 1, ..., 1], \tag{5}$$

$$\text{scale} = [\text{absmax}\left(1, \frac{1}{\sqrt{K}} \sum_{d=1}^{l} s_{i_d,1} O_{i_d}\right), \cdots, \text{absmax}\left(1, \frac{1}{\sqrt{K}} \sum_{d=1}^{l} s_{i_d,K} O_{i_d}\right)], \tag{6}$$

where we assume normal tokens are filled up with 1. The effect of victims can be qualified with the equation:

$$\mathbf{x}_{smooth} = 1/\text{scale}, \mathbf{u} = \max(|\mathbf{x}_{smooth}|)/\text{RMS}(|\mathbf{x}_{smooth}|) \tag{7}$$

To measure the effect of victims in the actual scenario, we first collect the activations from LLaMA3-8B. For activations from the Down Projector, spike outliers are 1000x larger than the medium value, as shown in Figure 7, where outliers in a channel-wise manner are not overly large.

To analyze the effect of smoothing rotated spike outliers, we apply the Monte Carlo approach by generating a token from the Gaussian distribution and inserting spike outliers according to statistics of spike outliers, then rotating, smoothing, and calculating $u$ as shown in Figure 8. Rotated tokens with multiple outliers are up and down across channels due to the effect of offset and stack. On the other hand, we can stack rotated tokens to obtain a consistent large scale across channels. As shown in Figure 8, normal tokens after smoothing are mostly easy to quantify but contrary when two outlier tokens are exhibited in one activation. The reason is that two tokens cannot cover the whole channel, where more stacked tokens can lead to lower $u$. Notably, the case with only two outlier tokens is rare but could potentially trouble Rotated Runtime Smooth.

### C.2  ANALYSIS EXTENT OF OUTLIER REMOVAL FOR DIFFERENT METHOD

We conduct experiments on mainstream models with different outlier smoothing approaches to analyze smoothness integrally rather than simulating with manual spike outliers. Specifically, we collect activations with models evaluated by WikiText-2 and apply different smoothing approaches. To measure the level of outliers; we set $\mu = \text{absmax}(\mathbf{t})/\|\mathbf{t}\|_2$, where $\mathbf{t}$ denotes one token. Figure 9 shows the specific impact of approaches on outliers on different LLM's components. For QKV_Projector, UP_Projector, and Gate_project, the activations are channel-wise consistent; hence, Runtime Smooth outperforms rotation, where pure rotated activations are sub-smooth. For Down_Projector, the intermediate activations contain spike outliers due to SwiGLU (Shazeer, 2020) functions; hence, Runtime Smooth suffers from the effect of the victim and fails to smooth. Rotated Runtime Smooth solves two kinds of outliers and consistently outperforms other approaches.

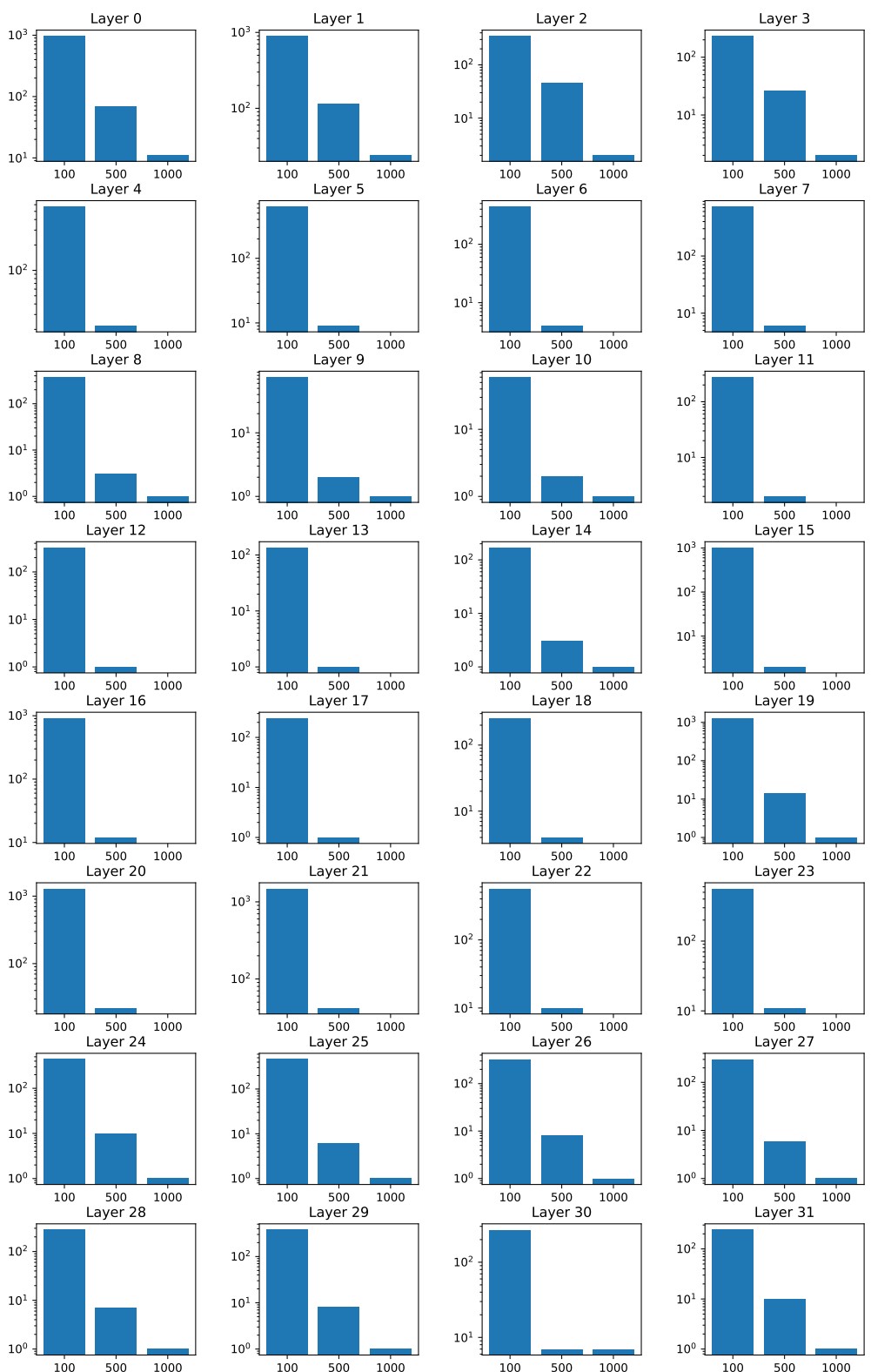

Figure 7: Collecting the activations as input of Down Projector with the full precision LLaMA3-8B model evaluating on WikiText-2 and counting the magnitude and number of spike outliers. The magnitude is calculated by $x/medium(t)$, where t is a token, and x is the element of the token. We separately count the spike outliers with different magnitude intervals.

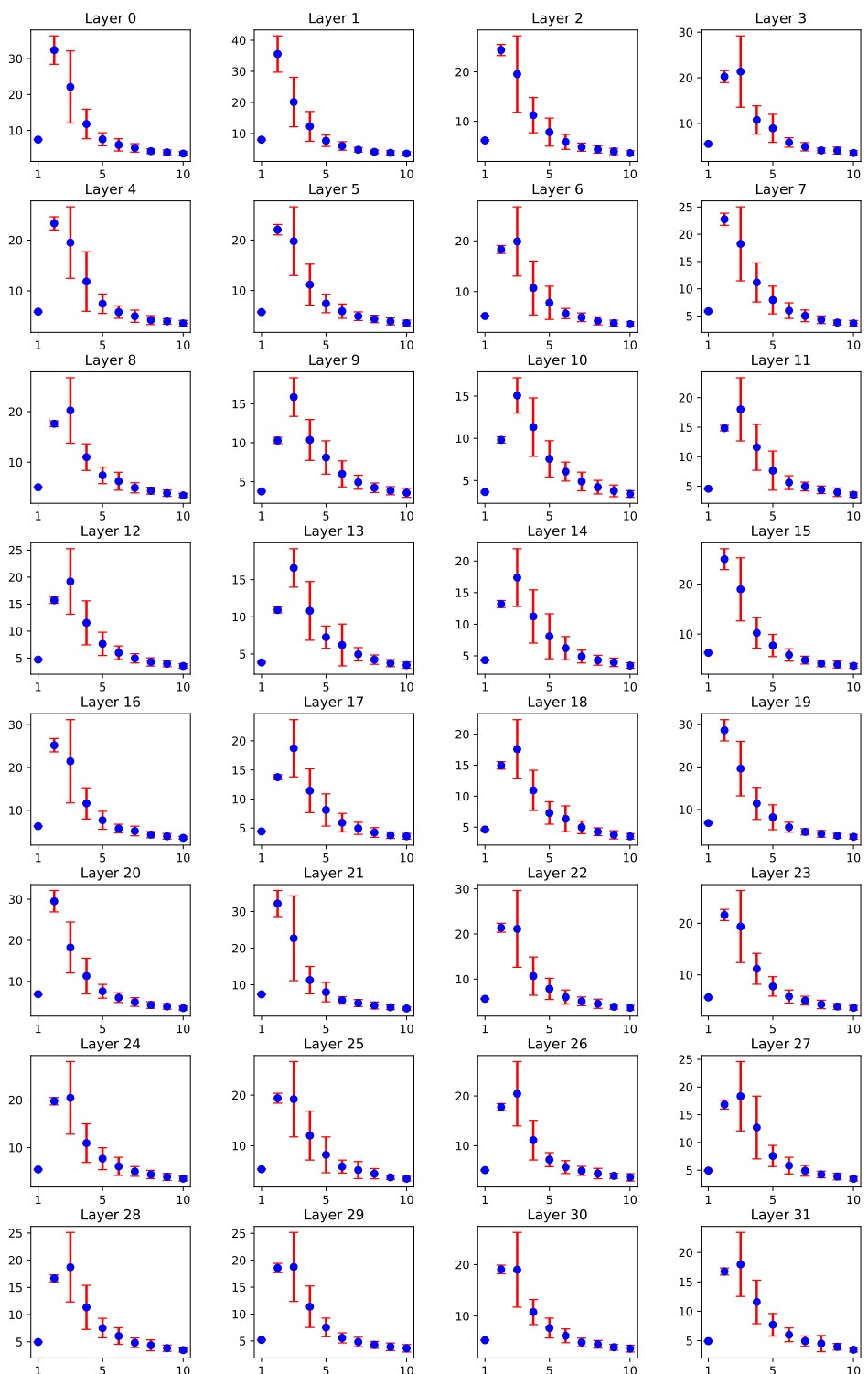

Figure 8: Simulation for the effect of victims after smoothing with rotated spike outliers. The magnitude and number of spike outliers are configured according to Figure 7. The X-axis denotes the number of spike tokens in an activation. The Y-axis denotes $u$ of the normal tokens divided by smooth scales.

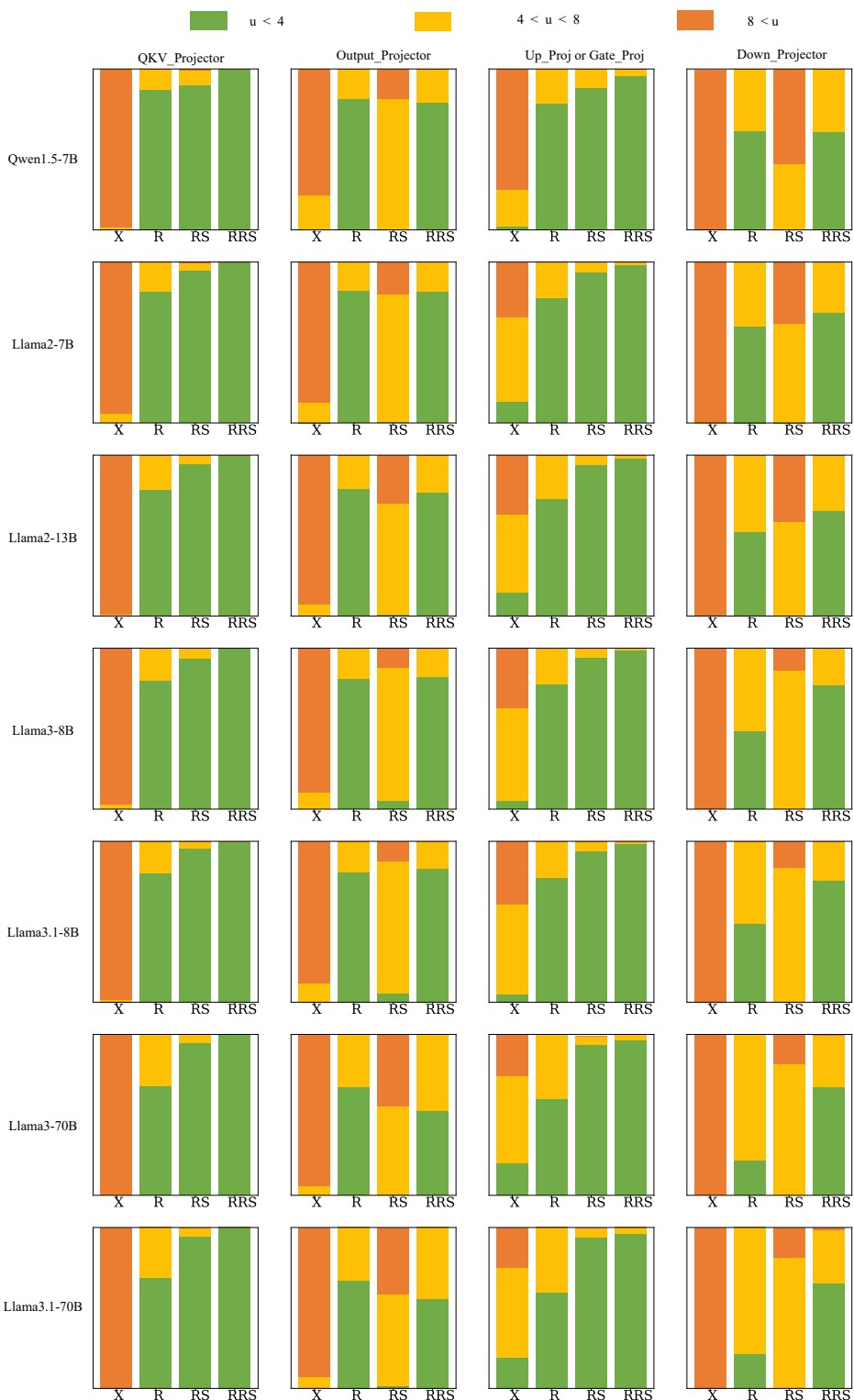

Figure 9: Statistic analysis of outlier removal with different smooth approaches. We collect the activations with full precision models evaluating on WikiText-2. 'X' denotes origin activations, 'R' denotes rotated activations, 'RS' denotes activations after Runtime Smooth, and 'RRS' denotes activations after Rotated Runtime Smooth.

