# OpenReview forum: "Rotated Runtime Smooth: Training-Free Activation Smoother for accurate INT4 inference"
_ICLR.cc/2025/Conference — ICLR 2025 Poster_

### Official Review · Reviewer_fKdX · 2024-10-30

**Soundness:** 3
**Presentation:** 2
**Contribution:** 2
**Rating:** 5
**Confidence:** 4

**Summary:**

This paper proposed a quantization technique designed to address outliers during Large Language Model (LLM) quantization, particularly for aggressive scenarios like W4A4 quantization. The method employs two steps: 1) mitigating channel-wise outliers through rotation, and 2) implementing online smoothing to further handle spike outliers. The authors demonstrate promising accuracy results across several benchmarks.

**Strengths:**

- The authors provide a comprehensive analysis of activation distribution patterns, offering well-reasoned solutions for each identified challenge.
- They proposed a training-free approach that outperforms existing training-based methods on some benchmarks, reducing engineering complexity and showing potential for practical applications in industry scenarios.

**Weaknesses:**

- Limited novelty: The core methodology is primarily based on existing approaches like [Smooth quantization](https://arxiv.org/abs/2211.10438 ) and [QuaRot](https://arxiv.org/abs/2404.00456 ), while the reordering technique shows overlap with existing research such as [ATOM](https://arxiv.org/abs/2310.19102) and [LLM.int8()](https://arxiv.org/abs/2208.07339).
- The experimental evaluation is limited in scope, focusing on common sense QA tasks. To better validate the method's effectiveness, the paper would benefit from:
    - More comprehensive comparisons with training-based methods like [SpinQuant](https://arxiv.org/abs/2405.16406) on more complex tasks, like [MMLU](https://arxiv.org/abs/2009.03300)
    - Detailed kernel performance comparisons with related research like [Qserve]( https://arxiv.org/abs/2405.04532), and [ATOM](https://arxiv.org/abs/2310.19102)

- Several aspects of the paper's presentation could be improved:
    - Figure 1: Show the challenges of existing methods, suggest adding visual explanations demonstrating how the proposed RRS technique addresses these challenges
   - Figure 1(b): Highlight the activation grouping methodology in the diagram
   - Figure 2(b): Provide a more detailed explanation of the probability metrics
   - Figure 6: Add clear y-axis labels and justify the unusual batch size choices (>1000).
   - Figure 9: Verify if the orange label represents `u > 8` instead of `u < 8`

**Questions:**

Please refer the last item of Weaknesses.

---

> ### Author Response · Authors · 2024-11-20
>
> Thanks for your comments and suggestions, which are valuable for enhancing our paper. The following are responses to individual concerns:
>
> **Novelty:**
> Previous works have untaclked problems for INT4 inference: 1. unmatched smoothing scales would fail for smoothing (SmoothQuant); 2. channel-wise smoothing would bring effect of victim (SmoothQuant); 3. rotated activations may still have channel-wise outliers and be sub-smooth for quantization (QuaRot). In this work, we employ Runtime Smooth to address the sub-smooth issue of rotation, and we also use a rotation technique to address the victim effect. We introduce a seamless and effective integration of the two techniques, which is non-rival.
>
> Moreover, the low efficiency of Runtime Smooth stems from its incompatibility with the hardware computation scheme, which only allows for the computation of one column of the matrix per computation unit. Therefore, we incorporate the reordering technique to collect outliers and normal values. Next, we apply the smoothing scale group-wise, which can approximate smoothness and align with the hardware computation scheme. The reordering technique in ATOM and LLM.int8() serves for mixed precision quantization, which share different motivations with us.
>
> **More downstream tasks:**
>
> | Model | Method | gsm8k_flexible (Math) | gsm8k_strict (Math) | MMLU | C-Eval (Chinese)|
> |--------------|------------|----------------|--------------|-------| ------|
> | Qwen2.5-7B | RRS (ours) | 70.96 | 62.85 | 63.00 | 66.6 |
> | Qwen2.5-7B | QuaRot | 62.77 | 52.99 | 62.15 | 66.1 |
> | Mistral | RRS (ours) | 37.30 | 36.69 | 56.65 | 40.6 |
> | Mistral | QuaRot | 31.15 | 30.62 | 55.16 | 39.3 |
> | LLaMA-3-8B | RRS (ours) | 31.46 | 31.38 | 53.51 | 35.5 |
> | LLaMA-3-8B | QuaRot | 24.71 | 24.56 | 50.85 | 31.3 |
> | LLaMA-3.1-8B | RRS (ours) | 36.08 | 35.93 | 54.96 | 38.8 |
> | LLaMA-3.1-8B | QuaRot | 28.50 | 28.27 | 51.64 | 34.1 |
>
>
> | Model | Method | HumanEval (code)|
> |--------------|------------|----------------|
> | Qwen2.5-7B | RRS (ours) | 60.98 |
> | Qwen2.5-7B | QuaRot | 49.39 |
>
> Limited by computation resources, more downstream tasks and models are under testing.
>
> **Kernel performance comparison:**
> We have carried out experiments to compare the performance of the kernel, as illustrated in Figure 4. ATOM uses the sub-channel A4W4 scheme, which brings more overhead compared with our method.
> For QServe, it only supports A8W4 and utilizes the A8W8 GEMM kernel. It is unfair to conduct comparisons between A4W4 and A8W8 since A4W4 lacks the support from hardware.
>
> We thank you for the presentation suggestion and will improve the presentation in the final version.

---

> > ### Comment · Reviewer_fKdX · 2024-11-23
> >
> > Figure 4 provides an illustration of the method but lacks detailed performance data. Specifically, it would be beneficial to include comparisons between your implemented kernel and float GEMM, as well as key metrics like end-to-end first token latency and decoding throughput.
> > Additionally, as far as I know, certain Nvidia GPUs, such as the A100 and RTX 3090, support W4A4 GEMM operation, please refer [1] for more details.
> >
> > 1. [PTX  `mma.m16n8k32` operation](https://docs.nvidia.com/cuda/parallel-thread-execution/#warp-level-matrix-fragment-mma-16832)

---

> > > ### Author Response · Authors · 2024-11-23
> > >
> > > An error occurred in the previous discussion. Actually, we have carried out experiments to compare the performance of the kernel, as illustrated in **Figure 6**. The kernel evaluation is based on NVBench.
> > > The result shown below validates that the proposed method only requires negligible overhead.
> > > ## A4W4 Per channel Quantization (QuaROT)
> > > | bsz  | hidden_dim |  CPU Time  |  GPU Time  |  Elem/s  |
> > > |------|------------|------------|------------|----------|
> > > |   16 |       4096 | 104.216 us |  90.830 us |   5.911T |
> > > |   32 |       4096 | 103.904 us |  91.001 us |  11.799T |
> > > |   64 |       4096 | 107.656 us |  93.930 us |  22.863T |
> > > |  128 |       4096 | 106.869 us |  93.981 us |  45.700T |
> > > |  256 |       4096 | 121.862 us | 109.621 us |  78.361T |
> > > |  512 |       4096 | 149.080 us | 135.876 us | 126.438T |
> > > | 1024 |       4096 | 188.206 us | 174.834 us | 196.528T |
> > > | 2048 |       4096 | 270.019 us | 255.065 us | 269.419T |
> > > | 4096 |       4096 | 432.151 us | 419.955 us | 327.271T |
> > >
> > > ## A4W4 Per channel Quantization + Runtime Smooth (Ours)
> > > | bsz  | hidden_dim |  CPU Time  |  GPU Time  |  Elem/s  |
> > > |------|------------|------------|------------|----------|
> > > |   16 |       4096 | 105.165 us |  93.437 us |   5.746T |
> > > |   32 |       4096 | 110.476 us |  96.535 us |  11.123T |
> > > |   64 |       4096 | 110.984 us |  97.432 us |  22.041T |
> > > |  128 |       4096 | 110.927 us |  95.742 us |  44.860T |
> > > |  256 |       4096 | 129.905 us | 114.573 us |  74.973T |
> > > |  512 |       4096 | 151.253 us | 139.174 us | 123.442T |
> > > | 1024 |       4096 | 192.143 us | 179.820 us | 191.079T |
> > > | 2048 |       4096 | 270.809 us | 258.284 us | 266.061T |
> > > | 4096 |       4096 | 437.859 us | 424.716 us | 323.602T |
> > >
> > >
> > > ## A4W4 Sub channel Quantization (ATOM)
> > > | bsz  | hidden_dim |  CPU Time  |  GPU Time  |  Elem/s  |
> > > |------|------------|------------|------------|----------|
> > > |   16 |       4096 | 104.334 us |  92.876 us |   5.781T |
> > > |   32 |       4096 | 106.628 us |  93.434 us |  11.492T |
> > > |   64 |       4096 | 108.771 us |  96.223 us |  22.318T |
> > > |  128 |       4096 | 109.460 us |  96.358 us |  44.573T |
> > > |  256 |       4096 | 127.548 us | 115.727 us |  74.226T |
> > > |  512 |       4096 | 157.194 us | 143.706 us | 119.549T |
> > > | 1024 |       4096 | 202.100 us | 188.224 us | 182.547T |
> > > | 2048 |       4096 | 293.830 us | 279.653 us | 245.731T |
> > > | 4096 |       4096 | 493.411 us | 475.318 us | 289.152T |

---

### Official Review · Reviewer_MyVY · 2024-11-04

**Soundness:** 3
**Presentation:** 3
**Contribution:** 2
**Rating:** 8
**Confidence:** 3

**Summary:**

The authors were the first to highlight that channel-wise and spike outliers cause unnecessary distributional expansion in the quantization of large language models (LLMs).

The authors proposed a concise and effective technique called Rotated Runtime Smooth, which reduces these outliers to ensure that the activation distribution for LLM quantization remains compact.

The authors functionally explained, along with the methodology, how Runtime Smooth addresses channel-wise outliers and Rotated Smooth effectively handles spike outliers.

The authors demonstrated effective performance improvements across various LLM models using this approach.

**Strengths:**

The authors focused on addressing channel-wise and spike outliers for the first time, which highlights a novel problem-solving approach and demonstrates significant performance improvements.

The methodology of Rotated Runtime Smooth (RRS) is simple, intuitive, and provides a clear and straightforward explanation of how it addresses each issue effectively.

By making their code publicly available, the authors have supported further research and made a valuable contribution to the ICLR community.

**Weaknesses:**

The explanation of the methodology is insufficient: For example, in Figures 1 and 4, which aim to explain Runtime Smooth, it is difficult to gain a proper understanding. The main text also lacks adequate explanation of these figures.

For instance, in Figure 4, while the activation and weight reorder process for Runtime Smooth is presented, there isn’t enough explanation about why this process does not alter or distort the final result. Similarly, in Figure 1, it is unclear what the value of s represents (in relation to the equations in the text) and how exactly it reduces channel-wise outliers and effectively resolves the victim issue caused by spike outliers.

These examples highlight the need for a more detailed and clear description of the methodology, possibly in the appendix. This would help ensure that readers can thoroughly understand the method. While the high-level intuition is conveyed, the reproducibility of the method should be clearly delivered to readers through the paper itself, without relying on the availability of the code.

It is essential to discuss why this study is limited to INT4. The limitations and advantages should be clearly articulated. For example, there is no comparison with recent trending quantization techniques, such as One-bit LLM. A thorough investigation of related work is necessary, and efforts should be made, at least indirectly, to demonstrate the applicability and effectiveness of the proposed method in relation to these latest quantization techniques. https://arxiv.org/abs/2310.11453

======

Assuming that the above points are adequately addressed, I will start with a borderline or higher score. However, if it is determined that these points are not sufficiently reflected, or based on the overall opinions of other reviewers or AC, I will consider adjusting my final score.

**Questions:**

Please refer to my comment for Weaknesses section.

---

> ### Author Response · Authors · 2024-11-12
>
> We appreciate the recognition of our contributions. We are delighted to explain the questions and make corresponding adjustments in the final version.
>
> **The meaning of Figure 1**: We present Figure 1 to point out the problem not solved by the existing methods - SmoothQuant.
> (a) SmoothQuant relies on a pre-computed smoothing scale from the calibration set. It would fail when faced with out-of-distribution (OOD) samples, whose channel-wise maximum values cannot match the pre-computed ones.
>
> (b) Even when addressing the unmatched problem by computing the smoothing scale online, SmoothQuant still fails for INT4. The reason is that SmoothQuant has to migrate the outlier of activation to weights, making quantization for weight unbearable.
>
> (c) Addressing the above problems, we find that the channel-wise smoothing method is still bounded by the spike outliers. The spike outliers have relatively small elements within the same channel. Hence, applying channel-wise smoothing would make those elements zero and cause the loss of information.
>
> For “it is unclear what the value of s represents (in relation to the equations in the text) and how exactly it reduces channel-wise outliers”, we have explained it in lines 151–153. s denotes the smoothing scale. It migrates the channel-wise outlier from activation to weight, tending to balance the quantization difficulty between activation and weight. We will make it more clear in Figure 1 for better understanding.
>
> For “effectively resolves the victim issue caused by spike outliers.” We claim that the victim effect is natural for smoothing-based methods (lines 162–164). And it motivates us to integrate rotation operations to address the spike outliers. Thanks for your suggestions. It is valuable to clarify the relationship between the victim issue and smoothing-based methods in the final version.
>
> **Description of the reorder process and the meaning of Figure 4**: The reorder process is to rearrange the channels of the weight and activation correspondingly, where the final result is equivalent. Specifically, the operation is defined as:
>
> Y = X @ W^T = \sum_{i=1}^{K} X_i @ W_i^T
>
> where X_i \in R^{n \times 1} and W_i \in R^{m \times 1} represent the \( i \)-th channel of  X  and W , respectively.  K  denotes the total number of channels.
>
> The activations and weights are rearranged and represented as X'  and W'. Hence, given that X'_i = W'_i, the final result remains consistent. We will provide further clarification in the final version.
>
> For the pipeline of Runtime Smooth, we have discussed it in sections 3.1 and 3.2. And we will make it more clear in the final version.
>
> **Why not lower quantization precision**: It is a good question. Quantization can be applied to weights and activations for different purposes. For weight-only quantization, it is to reduce the model size and I/O cost. To further accelerate inference, we should apply quantization to both weights and activations and utilize low-precision multiplication kernels, e.g., 8-bit for Hopper and 4-bit for Blackwell. For lower precision activation, it suffers from a heavy accuracy drop and has no gain in acceleration since it is not supported by the hardware. Moreover, the lower precision multiplication kernels are faced with numerical overflow and underflow problems. Hence, we do not apply lower precision quantization to activation. We have discussed related information in Lines 36-40.
>
> For the mentioned low-bit quantization, e.g., one-bit LLM, they focus on weight-only quantization. Whereas our proposed method focuses on activation smoothing.
>
>
> Hope the above explanations are helpful. We are pleased to have further discussion and are expecting your positive feedback.

---

> > ### Comment · Reviewer_MyVY · 2024-11-21
> >
> > Dear Authors :
> > Thank you for your comment on "Why not lower quantization precision." However, it seems that your argument would benefit from further support through diverse experimental results, rather than relying solely on discussion. As I initially suggested, could you provide supporting experimental evidence to substantiate your claim? Whether highlighting limitations or advantages, it is crucial to clearly articulate the contribution of this research through comparative experiments with existing related quantization techniques (e.g., one-bit LLM, though it has different setup like weight-only quantization - Please ensure the experiments are conducted as much as possible under identical conditions by explicitly demonstrating the differences). Thank you.

---

> ### Author Response · Authors · 2024-11-22
>
> We appreciate your suggestions. Lower quantization precision necessitates heavy quantization-aware training, such as one-bit LLM, to ensure accuracy recovery. Most of these works serve for weight-only quantization. To make a fair comparison, we apply activation quantization on top of weight quantization. Based on the quantization-aware training checkpoint (Llama3-8B-1.58-100B-tokens) from one-bit LLM, we conduct activation-weight quantization and testing on WikiText PPL. The result is shown below.
>
>
> We apply Runtime Smooth, Rotation, and Rotated Runtime Smooth for activation smoothing. The first four rows validate the effectiveness of those validation methods, especially our proposed Rotated Runtime Smooth. Comparing the second row with the fifth row, it shows that quantization-aware training with only 1.58B fails to improve accuracy of  INT4 inference with origin checkpoint.  However, comparing the third row with the last row demonstrates that QAT effectively addresses lower precision (3-bit) inference. Although the current hardware does not support lower precision inference, it shows the potential to gain better accuracy with QAT. Moreover, though there are pioneers working on QAT with both activations and weights (BitNet a4.8)[https://arxiv.org/html/2411.04965v1], it is still worthy to explore the potential of QAT with a rotated weight for better accuracy on extremely low precision inference.
>
> **Weight-Acitvation Quantization on LLaMA3-8B-Instruct:**
> | Checkpoint       | WBits | ABits | No smooth | Runtime Smooth | Rotation | Rotated Runtime Smooth |
> |------------------|-------|-------|-----------|----------------|----------|------------------------|
> | 1.58-100B-tokens | 1.58  | 8     | 13.29     | 13.28          | 13.27    | 13.26                  |
> | 1.58-100B-tokens | 1.58  | 4     | 2.1e5     | 36.86          | 29.38    | 15.76                  |
> | 1.58-100B-tokens | 1.58  | 3     | 4.1e4     | 578.30         | 217.06   | 139.34                 |
> | 1.58-100B-tokens | 1.58  | 2     | 1.2e6     | 3.3e5          | 3.3e5    | 2.3e5                  |
> | Origin           | 4     | 4     | 775.65    | 165.51         | 12.18    | 11.16                  |
> | Origin           | 3     | 3     | NAN       | NAN            | 3.3e3    | 6.3e2                  |
>
>
> However, as initially claimed, QAT requires heavy retraining. To observe the performance of the proposed method without QAT under lower precision, we conduct experiments on LLaMA3-8B. The result in the following table shows that the proposed methods are easy to fail under lower precision.
>
> **Weight-Acitvation Quantization on LLaMA3-8B:**
> | Method | WBits | ABits | WikiText PPL | MMLU   |
> |--------|-------|-------|--------------|--------|
> | RRS    | 4     | 4     | 8.11         | 53.5   |
> | QuaRot | 4     | 4     | 8.38         | 50.9   |
> | RRS    | 3     | 3     | 9.4e2        | 25.7   |
> | QuaRot | 3     | 3     | 5.3e3        | 24.5   |
> | RRS    | 2     | 2     | 3.8e5        | 24.7   |
> | QuaRot | 2     | 2     | 1.5e6        | 23.7   |
> | RRS    | 1     | 1     | NAN          | 22.9   |
> | QuaRot | 1     | 1     | NAN          | 22.9   |
>
>
> **Weight-Only Quantization on LLaMA3-8B:**
> | WBits | ABits | WikiText PPL | MMLU   |
> |-------|-------|--------------|--------|
> | 16    | 16    | 6.13         | 62.0   |
> | 4     | 16    | 7.61         | 56.1   |
> | 3     | 16    | 55.21        | 23.2   |
> | 2     | 16    | 1.2e5        | 22.9   |
> | 1     | 16    | NAN          | 22.9   |
>
>
> In summary, due to heavy retraining and unstable performance, it is still inefficient to apply QAT. And we believe our proposed training-free activation smoothing methods can benefit the community. Hope the above information could help you and we are expecting your positive feedback.

---

> > ### Comment · Reviewer_MyVY · 2024-11-24
> >
> > Thank you for conducting additional experiments and presenting the results. I recommend that authors include these findings in the Appendix to better emphasize (clarify) the contributions and limitations of this study, along with the other points raised in the rebuttal discussion. Furthermore, I also encourage the authors to update the open-source code accordingly to enhance its impact on the community. I have adjusted my score accordingly. Thank you.

---

> > > ### Author Response · Authors · 2024-11-25
> > >
> > > We sincerely appreciate the reviewer's constructive feedback and will refine the paper as suggested. Thank you sincerely for the improved rating.

---

### Official Review · Reviewer_f1tA · 2024-11-04

**Soundness:** 3
**Presentation:** 3
**Contribution:** 3
**Rating:** 6
**Confidence:** 2

**Summary:**

This paper extends SmoothQuant [Xiao 2023] by:
 - Performing Quip-like [Tseng 2024] rotation before activation smoothing;
 - Obtaining smoothing scale in runtime and not merging it into weights;

**Strengths:**

The paper is generally well-written, with a comprehensive ablation study (RRS vs RS and RRS vs SmoothQuant) demonstrating the effectiveness, which is well motivated by the preliminary section.

**Weaknesses:**

The paper lacks an evaluation on the clock-time (instead of the operation count) overhead introduced by the method. This is a concern given that the method determines activation smoothing scale during runtime.

**Questions:**

See Weaknesses

---

> ### Author Response · Authors · 2024-11-13
>
> Thanks for acknowledging our contribution! We are pleased to explain the overhead brought by Runtime Smooth.
>
> Section 4.5 has discussed the runtime overhead and Figure 6 shows that Runtime Smooth would bring negligible overhead compared with the original A4W4 per-channel setting. The 'Tops' on the y-axis represents the elements processed each second, which can reflect the runtime overhead.
>
> The complete efficiency metrics are listed below:
> ## A4W4 Per channel Quantization (QuaROT)
> | bsz  | hidden_dim |  CPU Time  |  GPU Time  |  Elem/s  |
> |------|------------|------------|------------|----------|
> |   16 |       4096 | 104.216 us |  90.830 us |   5.911T |
> |   32 |       4096 | 103.904 us |  91.001 us |  11.799T |
> |   64 |       4096 | 107.656 us |  93.930 us |  22.863T |
> |  128 |       4096 | 106.869 us |  93.981 us |  45.700T |
> |  256 |       4096 | 121.862 us | 109.621 us |  78.361T |
> |  512 |       4096 | 149.080 us | 135.876 us | 126.438T |
> | 1024 |       4096 | 188.206 us | 174.834 us | 196.528T |
> | 2048 |       4096 | 270.019 us | 255.065 us | 269.419T |
> | 4096 |       4096 | 432.151 us | 419.955 us | 327.271T |
>
> ## A4W4 Per channel Quantization + Runtime Smooth (Ours)
> | bsz  | hidden_dim |  CPU Time  |  GPU Time  |  Elem/s  |
> |------|------------|------------|------------|----------|
> |   16 |       4096 | 105.165 us |  93.437 us |   5.746T |
> |   32 |       4096 | 110.476 us |  96.535 us |  11.123T |
> |   64 |       4096 | 110.984 us |  97.432 us |  22.041T |
> |  128 |       4096 | 110.927 us |  95.742 us |  44.860T |
> |  256 |       4096 | 129.905 us | 114.573 us |  74.973T |
> |  512 |       4096 | 151.253 us | 139.174 us | 123.442T |
> | 1024 |       4096 | 192.143 us | 179.820 us | 191.079T |
> | 2048 |       4096 | 270.809 us | 258.284 us | 266.061T |
> | 4096 |       4096 | 437.859 us | 424.716 us | 323.602T |
>
>
> ## A4W4 Sub channel Quantization (ATOM)
> | bsz  | hidden_dim |  CPU Time  |  GPU Time  |  Elem/s  |
> |------|------------|------------|------------|----------|
> |   16 |       4096 | 104.334 us |  92.876 us |   5.781T |
> |   32 |       4096 | 106.628 us |  93.434 us |  11.492T |
> |   64 |       4096 | 108.771 us |  96.223 us |  22.318T |
> |  128 |       4096 | 109.460 us |  96.358 us |  44.573T |
> |  256 |       4096 | 127.548 us | 115.727 us |  74.226T |
> |  512 |       4096 | 157.194 us | 143.706 us | 119.549T |
> | 1024 |       4096 | 202.100 us | 188.224 us | 182.547T |
> | 2048 |       4096 | 293.830 us | 279.653 us | 245.731T |
> | 4096 |       4096 | 493.411 us | 475.318 us | 289.152T |
>
> Hope the above explanations are helpful. We are pleased to have further discussion and are expecting your positive feedback.

---

### Official Review · Reviewer_2Usn · 2024-11-04

**Soundness:** 2
**Presentation:** 2
**Contribution:** 2
**Rating:** 5
**Confidence:** 5

**Summary:**

The paper proposes a quantization method for LLMs, utilizing a smoothing operation during runtime to mitigate the issue of spike outliers and normalize values. Experiments conducted under INT4 settings validate the effectiveness of this method.

**Strengths:**

1. The paper is clearly structured, making it easy to follow.
2. The method’s motivation is substantiated by experimental results, lending it credibility. In particular, the experiments in Section 4.5 demonstrate the minimal overhead introduced by the method, highlighting its practicality.

**Weaknesses:**

1. **Figure 1 is confusing and hard to interpret.** The differences between panels (a) and (b) are unclear, as they are on completely different scales despite both sharing the same $ W $ and $ X $. Additionally, have you quantized $ \hat{X} = X \text{diag}(s)^{-1} $? The elements in $ \hat{x} $ appear to be integers, though they should be floats.

2. **Some terms and phrases are difficult to understand.** For instance, the caption for Figure 2(c) uses the phrase "channel-wise consistency" without providing any explanation, making it hard to grasp. Similarly, in point (1), what exactly is meant by "unmatched scale" in Figure 1?

3. **The motivation closely resembles prior work.** For example, DuQuant [1] also begins by addressing layers with massive outliers (in this paper, it is named as "spike") and tackles both massive and normal layers. Similarly, the method in [2] addresses large activation outliers and provides more analysis on this topic than the present paper.

4. **Some claims lack theoretical or experimental support.** For instance, the claim that "smoothing scales depending on the calibration set are prone to being unmatched with online activations" would benefit from experimental results demonstrating the likelihood of such a mismatch.

5. **Some explanations are too brief.** For example, the term "reorder" is mentioned without much elaboration, described only as "reordering the activations and weights according to the magnitude of smoothing scales."

6. **The method lacks clarity in its description.** In Equation 3, $ \hat{X} $ and $ \hat{W} $ appear to be in INT4 format, while $ s_i $ is a float (assuming this is correct, as the authors do not specify). How are these terms compatible for multiplication? A DeQuantize operation may need to be included somewhere in Equation 4.

7. **The ablation study feels incomplete.** The method employs several components—reordering, online smoothing, etc.—but the impact of each isn’t fully explored.

8. **The performance is underwhelming.** The proposed method does not appear to outperform SpinQuant [3] or DuQuant [1], even though it uses online quantization, whereas SpinQuant and DuQuant are offline methods.

Minors:
1. In Equation 4, $ c_{i,j} $ should be $ s_{i,j} $.

[1] Lin, H., Xu, H., Wu, Y., Cui, J., Zhang, Y., Mou, L., ... & Wei, Y. (2024). DuQuant: Distributing outliers via dual transformation makes stronger quantized LLMs. *arXiv preprint arXiv:2406.01721*.

[2] Yang, J., Kim, H., & Kim, Y. (2024). Mitigating Quantization Errors Due to Activation Spikes in GLU-Based LLMs. *arXiv preprint arXiv:2405.14428*.

[3] Liu, Z., Zhao, C., Fedorov, I., Soran, B., Choudhary, D., Krishnamoorthi, R., ... & Blankevoort, T. (2024). SpinQuant—LLM quantization with learned rotations. *arXiv preprint arXiv:2405.16406*.

**Questions:**

See Weaknesses.

---

> ### Author Response · Authors · 2024-11-20
>
> We appreciate your comments and suggestions that truly enhanced the quality of our paper.
>
> **Meaning of Figure 1 (W1, W2):**
> We present Figure 1 to highlight the issue that the existing methods fail to solve: SmoothQuant. SmoothQuant relies on a pre-computed smoothing scale that originates from the calibration set. It would fail when faced with out-of-distribution (OOD) samples, whose channel-wise maximum values cannot match the pre-computed ones.
>
> (b) SmoothQuant still fails for INT4, even when the unmatched problem is addressed by computing the smoothing scale online. The reason for this failure is that SmoothQuant must convert the activation outlier into weights, which makes the weight quantization process extremely challenging.
>
> (c) Addressing the above problems, we find that the channel-wise smoothing method is still bound by the spike outliers. The spike outliers have relatively small elements within the same channel. Hence, applying channel-wise smoothing would make those elements zero and cause the loss of information.
>
> Specifically, the difference between (a) and (b) comes from using the different smoothing scales. The former (SmoothQuant) uses the pre-computed one from the calibration dataset, which could be unmatched with the OOD samples. The latter computes the smoothing scale online and solves the unmatched problem. However, even when solving the unmatched problem, SmoothQuant still fails INT4 due to harder quantization for weights, which is the core idea of (b).
>
> **Motivations (W3):** We share the same motivation to address outliers during quantization as DuQuant, QuaRot, and SpinQuant do. However, we have observed that the rotated activations may still contain channel-wise outliers and remain sub-smooth for quantization purposes. Moreover, the rotation technique can assist in mitigating the victim effect, which poses a significant threat to the channel-wise smoothing method. Based on these two factors, we propose the Rotated Runtime Smooth method.
>
> **Experimental support for claims (W4):** Good point! We have validated the importance of matched scales as shown in Figure 3. However, it appears that we neglected to demonstrate the ease of unmatching the fixed smooth scales from the calibration set. To validate the fact, we reparameterize the layer norm and weights according to a fixed smooth scale from the calibration set. Then we collect the activations from WikiText-2-Raw-V1. If the activation matches the smooth scale, u = max(|t|)/RMS(t) tends to be 1. For comparison, we also collect activations without reparameterizing.
>
> | Method | u < 4 | 4 < u < 8 | 8 < u | PPL |
> |--------|-------|----------|-------|-----|
> | SmoothQuant | 51.9% | 24.6% | 23.4% | 370.612 |
> | origin | 11.0% | 54.2% | 33.3% | 389.865 |
> | Runtime Smooth | 100% | 0 | 0 | 11.943 |
>
> The results validate that fixed runtime smooth scales are easy to unmatch, which leads to performance drops.
>
>
> **Provide further explanations for the terms (W2, W5, and W6):** Thanks for your suggestions. We will add more explanations for the terms in the paper.
>
> **Incomplete ablation study (W7):** Here we detail the components. including 1. online smoothing, 2. reordering, and 3. a rotation technique. The reordering serves for the online smoothing from the perspective of efficiency; hence, it could not stand alone. The possible combinations are as follows: 1. (1); 2. (1); 3. (1+2); 4. (3); 5. (1+3); 6. (1+2+3). 1. denotes GPTQ (Table 1); 2. denotes Runtime Smooth (Table 1); 3. denotes Runtime Smooth with reordering (Table 4); 4. denotes QuaRot (Table 1); 5. denotes Rotated Runtime Smooth (Table 4); 6. indicates Rotated Runtime Smooth with reordering (Table 1). Hence, the ablation study is complete.
>
> **Additional comparisons with existing methods (W8):** The results show that our method outperforms the existing methods, and their variants featured with Runtime Smooth can also achieve better performance.
> | Method | gsm8k_flexible | gsm8k_strict | MMLU | arc_challenge | arc_easy | boolq | openbookqa | average |
> |--------------|----------------|--------------|------|---------------|----------|-------|------------|---------|
> | RRS | 31.4 | 31.3 | 53.5 | 44.8 | 71.1 | 73.6 | 42.4 | 57.9 |
> | DuQuant | 24.4 | 24.0 | 49.4 | 45.8 | 68.7 | 71.3 | 42.2 | 57.0 |
> | DuQuant+RS | 32.9 | 32.4 | 55.7 | 47.5 | 74.0 | 76.2 | 42.6 | 60.1 |
> | SpinQuant+RS | 23.1 | 23.2 | 52.2 | 45.1 | 70.6 | 73.6 | 39.4 | 57.1 |
> | SpinQuant | 22.9 | 22.6 | 48.4 | 43.2 | 67.3 | 72.7 | 38.8 | 55.5 |

---

> ### Author Response · Authors · 2024-12-02
>
> Dear reviewer. As the rebuttal deadline is approaching, I want to kindly mention that you have not responded to my rebuttal. Timely feedback from yours would be constructive in preparing a comprehensive revision. ﻿ I understand everyone has a busy schedule, and I truly appreciate your efforts. Thank you for your time and support. ﻿ Best regards.

---

> > ### Comment · Reviewer_2Usn · 2024-12-03
> > **Response**
> >
> > Thank you for the authors’ detailed response. After the rebuttal, I find that most of the concerns have been addressed, and the presentation has improved significantly. However, there are still a couple of issues: (1) the method still seems somewhat lacking in novelty, as also pointed out by reviewer fKdX, and the response does not fully address my concerns in this regard; (2) more insights into the benefits of runtime could further strengthen the paper. The response to W4 focuses mainly on empirical results without offering deeper insights. It would be helpful if there were observations explaining the differences in results between online and offline operations. Additionally, while Figure 1 is a useful toy example, providing some statistical results would better illustrate these points, if possible.
> >
> > Overall, though, most of my concerns have been addressed, so I am raising my score from 3 to 5.

---

> > > ### Author Response · Authors · 2024-12-03
> > >
> > > We sincerely appreciate the improved rating, and we are willing to have a further discussion.
> > >
> > > **About novelty:** Previous works have tackled problems for INT4 inference: 1. unmatched smoothing scales would fail for smoothing (SmoothQuant); 2. channel-wise smoothing would bring the effect of the victim (SmoothQuant); 3. rotated activations may still have channel-wise outliers and be sub-smooth for quantization (QuaRot). In this work, we employ Runtime Smooth to address the sub-smooth issue of rotation, and we also use a rotation technique to address the victim effect. We introduce a seamless and effective integration of the two techniques, which is non-rival.
> > >
> > >
> > > Moreover, we would like to emphasize that making Runtime Smooth efficient is nontrivial because it is incompatible with the hardware computation scheme. Therefore, we incorporate the reordering technique to collect outliers and normal values. Next, we apply the smoothing scale group-wise, which can approximate smoothness and align with the hardware computation scheme.
> > >
> > > We have sent the above content to reviewer fKdX but received no response. On the other hand, our novelty has been acknowledged by the reviewer MyVY.
> > >
> > > **About more insights for the success of Runtime Smooth:**  The **theoretical support** of Channel-Wise Smooth is that "Outliers persist in the fixed channels," which is a common understanding in the previous works (SmoothQuant, Qserve). Applying channel-wise smooth scales is effective in this situation because division eliminates the outliers.
> > > However, from our observation of activations, outliers not only persist in the fixed channels but also exist in the dynamic channels. In other words, outliers exist channel-wise but not only in the fixed channels. Hence, previous works using fixed smooth scales might fail for smoothing due to the mismatch. We argue that the key to the success of Runtime Smooth is that it can effectively collect outliers and normal values, which is the most important contribution of Runtime Smooth.
> > >
> > > The **experiment** validates the concept of Runtime Smooth, where a smaller u indicates a smoother activation. The table illustrates that offline smoothing does not work for smoothing purposes.
> > > | Method | u < 4 | 4 < u < 8 | 8 < u | PPL |
> > > |--------|-------|----------|-------|-----|
> > > | SmoothQuant | 51.9% | 24.6% | 23.4% | 370.612 |
> > > | origin | 11.0% | 54.2% | 33.3% | 389.865 |
> > > | Runtime Smooth | 100% | 0 | 0 | 11.943 |
> > >
> > > Moreover, we would like to emphasize that the benefit of Runtime Smooth is obvious. As we previously discussed, achieving efficiency in Runtime Smooth is more challenging.
> > >
> > > Again, thank you for your time and constructive comments. We hope the above content can fully address your concerns.

---

### Official Review · Reviewer_fXyG · 2024-11-04

**Soundness:** 3
**Presentation:** 2
**Contribution:** 2
**Rating:** 5
**Confidence:** 2

**Summary:**

A plug-and-play quantization method based on runtime smoothing and the rotation operation of activations has been proposed. Runtime smoothing is responsible for eliminating channel-wise outliers, while the rotation operation mitigate the gap between spike outliers and normal values, resulted by channel-wise smoothing. Experimenting with INT4 inference on different LLMs shows that the proposed method can reduce the perplexity compared to other approaches.

**Strengths:**

- Combining two exiting ideas in the Quantization literature to overcome both channel-wise and spikes outliers
- Comprehensive experiments with 3 different LLMs, including LLaMA, Qwen, Mixtral and Mistral models on WikiText-2 perplexity

**Weaknesses:**

Some notations need to be corrected. For example,
- Line 168: identity matrix and number 1 should be distinguished. $|.|$ the norm needs to be determined.
- Line 175: $absmax$ ---- > $abs(\max)$
- Equation (1): $\mathrm{X_j}$ needs to be defined as the columns of matrix $\mathrm{X}$.
- Equation(2): $/$ is not a valid operation for matrices. You need to write the formula as a multiplication with a diagonal matrix.
- line 214: The condition needs to be re-written: $s_j$ cannot be taken out of the sum.

Another issue is that the proposed method has been only tested for the  WikiText-2 perplexity. Can the approach is applicable for other modalities and tasks. For instance, ASR, speech translation, Image understanding, etc.

There is only a small improvement over QuaRot approach. I am wondering if your results are statistical significant?

**Questions:**

Please see my comments above.

---

> ### Author Response · Authors · 2024-11-20
>
> Thanks for your valuable suggestions, which help us enhance the clarity of this paper and make it more understandable for the wider community. We thoroughly examined your suggestions and our manuscript to rectify any typos, ambiguous descriptions, and unclear settings in the revised version.
>
> We have tested methods on WikiText-2 PPL (Table 1) and Common Sense QA (Table 2). To address performance concerns, we have incorporated additional downstream tasks such as MMLU, GSM8k (Math), CEVAL (Chinese), HumanEval (code). The results below indicate that our method surpasses QuaRot in all tasks. The non-trival improvement on GSM8k demonstrates the effectiveness of runtime smoothing for data from diverse domains.
> | Model | Method | gsm8k_flexible (Math) | gsm8k_strict (Math) | MMLU | C-Eval (Chinese)|
> |--------------|------------|----------------|--------------|-------| ------|
> | Qwen2.5-7B | RRS (ours) | 70.96 | 62.85 | 63.00 | 66.6 |
> | Qwen2.5-7B | QuaRot | 62.77 | 52.99 | 62.15 | 66.1 |
> | Mistral | RRS (ours) | 37.30 | 36.69 | 56.65 | 40.6 |
> | Mistral | QuaRot | 31.15 | 30.62 | 55.16 | 39.3 |
> | LLaMA-3-8B | RRS (ours) | 31.46 | 31.38 | 53.51 | 35.5 |
> | LLaMA-3-8B | QuaRot | 24.71 | 24.56 | 50.85 | 31.3 |
> | LLaMA-3.1-8B | RRS (ours) | 36.08 | 35.93 | 54.96 | 38.8 |
> | LLaMA-3.1-8B | QuaRot | 28.50 | 28.27 | 51.64 | 34.1 |
>
>
> | Model | Method | HumanEval (code)|
> |--------------|------------|----------------|
> | Qwen2.5-7B | RRS (ours) | 60.98 |
> | Qwen2.5-7B | QuaRot | 49.39 |
>
> Limited by computation resources, more downstream tasks and models are under testing.
>
>
> We have also included additional baselines to ensure a fair comparison. Please refer to the global rebuttal for further details.
>
> Moreover, we appreciate the suggestions for testing on other modalities. We will consider this in our future work.

---

> ### Author Response · Authors · 2024-11-30
>
> Dear reviewer. As the rebuttal deadline is approaching, I want to kindly mention that you have not responded to my rebuttal. Timely feedback from yours would be constructive in preparing a comprehensive revision. ﻿ I understand everyone has a busy schedule, and I truly appreciate your efforts. Thank you for your time and support. ﻿ Best regards.

---

### Author Response · Authors · 2024-11-20

Global response:
We are cheerful that the reviewers found our method well motivated (Reviewer f1tA), novel (Reviewer MyVY), and well presented (all reviewers). We extend our gratitude to the reviewers for their constructive feedback. We have addressed all the concerns raised by the reviewers in the revised manuscript. We have made the following changes in the revised manuscript:

A major concern most reviewers share is the absence of a wider variety of downstream tasks and baselines. In the previous version, we conducted comparisons between our methods and those of SmoothQuant, QuaRot on WikiText PPL, and Common Sense QA. In the revised version, we add more downstream tasks (MMLU, GSM8k (Math), CEVAL (Chinese), HumanEval (code)) and baselines (SpinQuant and DuQuant). It is important to note that our method provides an effective and efficient plug-and-play activation smoother, and we will continue to investigate the performance of future approaches that incorporate our method.

**More downstream tasks**

We evaluate our method on two more downstream tasks: MMLU and GSM8k, with 7/8B models from Qwen2.5, Mistral, LLaMA 3, and LLaMA 3.1. The results show that our method effectively smooths the activations and is robust to out-of-distribution (OOD) data, e.g., math and code, thereby achieving non-trivial improvements over the baselines.


| Model | Method | gsm8k_flexible (Math) | gsm8k_strict (Math) | MMLU | C-Eval (Chinese)|
|--------------|------------|----------------|--------------|-------| ------|
| Qwen2.5-7B | RRS (ours) | 70.96 | 62.85 | 63.00 | 66.6 |
| Qwen2.5-7B | QuaRot | 62.77 | 52.99 | 62.15 | 66.1 |
| Mistral | RRS (ours) | 37.30 | 36.69 | 56.65 | 40.6 |
| Mistral | QuaRot | 31.15 | 30.62 | 55.16 | 39.3 |
| LLaMA-3-8B | RRS (ours) | 31.46 | 31.38 | 53.51 | 35.5 |
| LLaMA-3-8B | QuaRot | 24.71 | 24.56 | 50.85 | 31.3 |
| LLaMA-3.1-8B | RRS (ours) | 36.08 | 35.93 | 54.96 | 38.8 |
| LLaMA-3.1-8B | QuaRot | 28.50 | 28.27 | 51.64 | 34.1 |


| Model | Method | HumanEval (code)|
|--------------|------------|--------|
| Qwen2.5-7B | RRS (ours) | 60.98 |
| Qwen2.5-7B | QuaRot | 49.39 |

Limited by computation resources, more downstream tasks and models are under testing.


**More baseline**

We compare our method with two more baselines: SpinQuant and DuQuant and their variants featured with Runtime Smooth (RS). The results show that our Runtime Smooth is well generalisable and valid, demonstrating the importance of runtime smoothing for various data from different domains.

| Method | gsm8k_flexible | gsm8k_strict | MMLU | arc_challenge | arc_easy | boolq | openbookqa | average |
|--------------|----------------|--------------|------|---------------|----------|-------|------------|---------|
| RRS | 31.4 | 31.3 | 53.5 | 44.8 | 71.1 | 73.6 | 42.4 | 57.9 |
| DuQuant | 24.4 | 24.0 | 49.4 | 45.8 | 68.7 | 71.3 | 42.2 | 57.0 |
| DuQuant+RS | 32.9 | 32.4 | 55.7 | 47.5 | 74.0 | 76.2 | 42.6 | 60.1 |
| SpinQuant+RS | 23.1 | 23.2 | 52.2 | 45.1 | 70.6 | 73.6 | 39.4 | 57.1 |
| SpinQuant | 22.9 | 22.6 | 48.4 | 43.2 | 67.3 | 72.7 | 38.8 | 55.5 |


Another concern is the clarity of the paper. We thank the reviewers for their feedback. We are actively addressing the concerns raised and refining our paper in the revised version to enhance its understanding for the wider community. Specifically;

1. We fix notations and typos in the paper. (R1, R2, R5)
2. We have included more related work for clarity. (R2, R5)
3. Provide more details in the caption for Figure 1. (R2, R4)
4. Clarify the process of reordering the caption in Figure 4. (R2, R4)
5. Clarify the detail of probability metrics in Figure 2(b). (R5)

Finally, we would like to express our great appreciation and excitement that several reviewers have recognized our  potential. We want to emphasize that our work contributes to the community not only by proposing a particular approach that works well with A4W4 inference, but also by providing a lightweight, robust activation smoother plugin. We believe that it is worth publishing to stimulate further discussion.

---

### Meta-Review · Area_Chair_gN4t · 2024-12-22

**Metareview:**

Large language models achieve impressive performance as their scale increases but incur high computational and memory costs. Quantization mitigates these costs but struggles with activation outliers, hindering INT4 quantization. This work introduces Rotated Runtime Smooth (RRS), a plug-and-play activation smoother combining Runtime Smooth (RS) to handle channel-wise outliers and a Rotation operation to address spike outliers. RRS outperforms state-of-the-art methods, achieving significant improvements, including reducing WikiText-2 perplexity from 57.33 to 6.66 in INT4 inference.


Summary of the strengths and weaknesses of the paper according to the reviewers:

Strengths:

+The paper introduces a plug-and-play quantization method combining runtime smoothing and rotation to address channel-wise and spike outliers.

+This paper claims negligible overhead compared to existing quantization methods while achieving better perplexity scores.

+It also extends experimental evaluation to multiple downstream tasks and baselines, demonstrating robustness across different domains.

+Finally, the reviewers appreciated the clear structure and logical flow, that they thought was helpful in understanding the methodology.

Weaknesses:

-Limited Novelty: Several reviewers noted overlaps with prior work (e.g., SmoothQuant, QuaRot, ATOM), questioning the methodological originality.

-Incomplete Methodology Explanation: Some figures (e.g., Figures 1 and 4) and terms were inadequately explained, making parts of the paper challenging to follow.

-Lack of Diversity in Evaluation: Initially focused on a narrow set of tasks (e.g., WikiText-2), prompting concerns about the generalizability of the results.

-Minor Performance Gains: Marginal improvements over prior methods led to skepticism about the significance of the results.

In their rebuttal, the authors addressed reviewer concerns by expanding experiments to additional tasks (e.g., MMLU, GSM8k) and demonstrating consistent improvements across diverse domains. They provided detailed runtime efficiency metrics, showing negligible overhead for their method. The authors clarified the novelty of their approach, emphasizing how the integration of Runtime Smooth and rotation resolves issues with prior methods. They also explained unclear figures and terms, committing to improving the final manuscript for better clarity. These efforts successfully convinced some reviewers to raise their scores, though skepticism about novelty and deeper theoretical insights persisted.

Reviewer reactions to the rebuttal were mixed. Some reviewers appreciated the additional experiments and detailed clarifications, leading to improved scores (e.g., Reviewer MyVY). They acknowledged the authors' efforts in addressing concerns about evaluation scope, efficiency, and clarity. However, other reviewers (e.g., Reviewer 2Usn) remained skeptical about the novelty of the approach and the lack of deeper theoretical insights, with concerns about incremental contributions persisting. The paper is a bit borderline with polarizing reviews. But I do think the authors response clarified major issues. While other issues persist I think the ICLR community will benefit from the discussion and I thus recommend acceptance. That said, I recommend the authors address the remaining issues raised by the reviewers including novelty concerns.

**Additional Comments On Reviewer Discussion:**

As mentioned above, reviewer reactions to the rebuttal were mixed. Some reviewers appreciated the additional experiments and detailed clarifications, leading to improved scores (e.g., Reviewer MyVY). They acknowledged the authors' efforts in addressing concerns about evaluation scope, efficiency, and clarity. However, other reviewers (e.g., Reviewer 2Usn) remained skeptical about the novelty of the approach and the lack of deeper theoretical insights, with concerns about incremental contributions persisting.

---

### Decision · Program_Chairs · 2025-01-22

Accept (Poster)